# VECO: Variable Encoder-decoder Pre-training for Cross-lingual Understanding and Generation

## Abstract

Recent studies about learning multilingual representations have achieved significant performance gains across a wide range of downstream cross-lingual tasks. They train either an encoder-only Transformer mainly for understanding tasks, or an encoder-decoder Transformer specifically for generation tasks, ignoring the correlation between the two tasks and frameworks. In contrast, this paper presents a variable encoder-decoder (VECO) pre-training approach to unify the two mainstreams in both model architectures and pre-training tasks. VECO splits the standard Transformer block into several sub-modules trained with both inner-sequence and cross-sequence masked language modeling, and correspondingly reorganizes certain sub-modules for understanding and generation tasks during inference. Such a workflow not only ensures to train the most streamlined parameters necessary for two kinds of tasks, but also enables them to boost each other via sharing common sub-modules. As a result, VECO delivers new state-of-the-art results on various cross-lingual understanding tasks of the XTREME benchmark covering text classification, sequence labeling, question answering, and sentence retrieval. For generation tasks, VECO also outperforms all existing cross-lingual models and state-of-the-art Transformer variants on WMT14 English-to-German and English-to-French translation datasets, with gains of up to 1~2 BLEU.

## 1 Introduction

Driven by the striking success of pre-trained language models (Devlin et al., 2019), recent cross-lingual pre-training (Lample & Conneau, 2019; Liu et al., 2020b) has attracted an increasing of attention. It provides cross-lingual contextualized representations for the inputs of different languages, which significantly advances performance in both natural language understanding (NLU) and generation (NLG) tasks.

There are two mainstream architectures in current cross-lingual pre-training literature: *encoder-only* and *encoder-decoder*. The former like XLM (Lample & Conneau, 2019) focuses on conducting masked language modeling (MLM) with a single Transformer (Vaswani et al., 2017) encoder. This paradigm is naturally compatible with various NLU tasks, but tends to suffer from limited gains on cross-lingual generation tasks (e.g., machine translation) due to the lack of effective decoder initialization. In contrast, the latter like mBART (Liu et al., 2020b) attempts to pre-train the encoder-decoder Transformer via denoising auto-encoding tasks to provide complete initialization for downstream generation tasks. However, when applied in NLU scenarios, it usually requires more computation and memory to match the performance of the encoder-only models.

In light of the above pros and cons, this work presents Variable Encoder-deCOder (VECO) pre-training, which targets at providing pre-trained model initialization for both the *encoder-only* and *encoder-decoder* Transformer with the most streamlined parameters. We observe that Transformer encoder and decoder blocks have two common modules: `SelfAttention` and `FFN` (feed-forward network), with the main difference that the latter introduces an extra `CrossAttention` (attention across from the encoder to the decoder) module. Inspired by the lottery ticket hypothesis (Frankle & Carbin, 2018), we split the standard Transformer block into three independent modules

---

*Equal contribution.

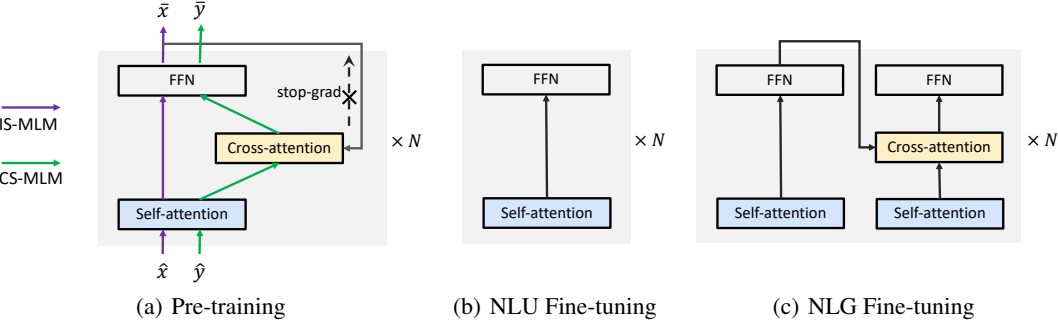

(a) Pre-training        (b) NLU Fine-tuning        (c) NLG Fine-tuning

Figure 1: The overview of VECO. During pre-training, we feed two masked segments $\hat{x}$ and $\hat{y}$ into different modules to perform inner-sentence mask language modeling (IS-MLM) and cross-sentence mask language modeling (CS-MLM). More specifically, the masked segment $\hat{x}$ can only attend to its context via self-attention to recover the original tokens $\bar{x}$ (IS-MLM), while masked segment $\hat{y}$ can attend to its preceding tokens via self-attention and the context $\hat{x}$ via cross-attention to predict the original tokens $\bar{y}$ (CS-MLM). For downstream NLU tasks, we throw out the cross-attention module and only fine-tune on the self-attention and FFN modules acted as an encoder. For NLG tasks, we keep all modules to initialize the corresponding encoder and decoders.

$\{\texttt{SelfAttention}, \texttt{CrossAttention}, \texttt{FFN}\}$ to be collaboratively trained via two specific MLM tasks. After that, we rebuild the desired complete architecture applicable for NLU or NLG with different specific combinations of these modules during fine-tuning.[1] Specifically, to be equipped with the ability of language understanding during pre-training, $\texttt{SelfAttention}$ and $\texttt{FFN}$ are assembled into a standard Transformer encoder for conducting inner-sequence masked language modeling (IS-MLM). In terms of generation, $\texttt{SelfAttention}$, $\texttt{CrossAttention}$, and $\texttt{FFN}$ act together as the decoder in the standard sequence-to-sequence model, and are trained by the elaborately designed cross-sequence masked language modeling (CS-MLM) task. When applied to downstream fine-tuning, both $\texttt{SelfAttention}$ and $\texttt{FFN}$ modules constitute the Transformer encoder for contextual modeling in NLU or NLG, or cooperate with additional $\texttt{CrossAttention}$ to provide the effective initialization of Transformer decoder. With such kind of workflow, VECO can be applied to both NLU and NLG tasks with the most streamlined parameters, which significantly reduces computational overhead and memory costs. Moreover, IS-MLM is specifically designed for understanding of individual sequences, while both understanding and generation tasks can benefit from CS-MLM. With such parameter sharing, VECO enables $\texttt{SelfAttention}$ and $\texttt{FFN}$ modules to be jointly trained by the two MLMs, which boosts both NLU and NLG performance.

We validate VECO on a variety of representative cross-lingual NLU and NLG benchmarks. For cross-lingual understanding tasks, we conduct experiments on the XTREME benchmark consisting of 9 cross-lingual tasks, including text classification, sequence labeling, question answering, and sentence retrieval. VECO ranks first at the XTREME leaderboard[2] at the submission deadline and obtains new state-of-the-art results on most of the tasks. For cross-lingual generation tasks, we validate VECO on the widely used WMT14 English-German and English-French machine translation benchmarks. VECO obtains 44.4 and 31.5 BLEU scores, consistently outperforming existing cross-lingual pre-training approaches and state-of-the-art Transformer variants by around 1∼2 BLEU.

## 2   VARIABLE ENCODER-DECODER PRE-TRAINING

### 2.1   BACKBONE NETWORK

The backbone network of VECO is composed of a stack of $N$ identical layers. Each layer has three modules, consisting of a *required* self-attention module, an *optional* cross-attention module, and a *required* feed-forward linear module. Both self-attention and cross-attention modules are based on

---

[1]Thus the word *variable* means that the backbone Transformer varies during pre-training and fine-tuning.
[2]https://sites.research.google/xtreme

the multi-head attention (Vaswani et al., 2017):

$$\texttt{MultiHead}(\mathbf{Q}, \mathbf{K}, \mathbf{V}) = \texttt{Concat}(\text{head}_1, ..., \text{head}_\text{h})\mathbf{W^O}$$
$$\text{head}_\text{i} = \texttt{Attention}(\mathbf{QW^Q}, \mathbf{KW^K}, \mathbf{VW^V}) \tag{1}$$

where $\mathbf{W^O}, \mathbf{W^Q}, \mathbf{W^K}$ and $\mathbf{W^V}$ are parameter matrices. $\texttt{Attention}(a, b, c)$ represents the attention operation with $a$ as query, $b$ as key, and $c$ as value. We refer the readers to Vaswani et al. (2017) for more details.

The main difference of the self-attention module and cross-attention module is that $\mathbf{Q} = \mathbf{K} = \mathbf{V}$ holds in the self-attention module while only $\mathbf{K} = \mathbf{V}$ exists in the cross-attention module. We formalize these two modules as:

$$\texttt{SelfAttention}(\mathbf{x}; \theta_\mathbf{s}) = \texttt{AddNorm}\big(\texttt{MultiHead}(\mathbf{Q} = \mathbf{x}, \mathbf{K} = \mathbf{x}, \mathbf{V} = \mathbf{x})\big)$$
$$\texttt{CrossAttention}(\mathbf{x}, \mathbf{y}; \theta_\mathbf{c}) = \texttt{AddNorm}\big(\texttt{MultiHead}(\mathbf{Q} = \mathbf{y}, \mathbf{K} = \mathbf{x}, \mathbf{V} = \mathbf{x})\big) \tag{2}$$

where $\theta_\mathbf{s}$ and $\theta_\mathbf{c}$ are the corresponding parameters, and $\texttt{AddNorm}$ denotes a residual connection (He et al., 2016) with a post layer normalization (Ba et al., 2016).

After that, a fully connected feed-forward network is applied to each element of input independently:

$$\texttt{FFN}(\mathbf{x}; \theta_\mathbf{f}) = \texttt{AddNorm}\big(\mathbf{W_2}\,\texttt{GeLU}\,(\mathbf{W_1 x})\big) \tag{3}$$

where $\theta_\mathbf{f} = \{\mathbf{W_1}; \mathbf{W_2}\}$ are matrices of parameters.

## 2.2 PRE-TRAINING OBJECTIVES

In cross-lingual pre-training scenarios, we can utilize both monolingual and bilingual data widely used in previous works (Lample & Conneau, 2019; Chi et al., 2020b; Yang et al., 2020). We formalize both the two adjacent segments in the monolingual corpus and a pair of parallel sentence in the bilingual corpus as $(\boldsymbol{x}, \boldsymbol{y})$. We firstly adopt the same mask strategy like BERT (Devlin et al., 2019) to construct the masked input $(\hat{\boldsymbol{x}}, \hat{\boldsymbol{y}})$. Then, the backbone Transformer takes the input to perform the inner-sequence and cross-sequence masked language modeling, which enables the model to be optimized jointly for cross-lingual language understanding and cross-sequence generation.

**IS-MLM: Inner-Sequence Masked Language Modeling** To be equipped with the ability of language understanding, we perform masked language modeling of a single-sequence on the self-attention and FFN modules, while skipping the cross-attention modules. As shown in Figure 1, the purple lines show the forward process of the IS-MLM task in each layer. To be specific, the embeddings of the masked input sequence $\hat{\boldsymbol{x}}$ are fed into the self-attention and FFN modules in each layer to get a contextual presentation $\mathbf{X}^{(i)}$:

$$\mathbf{H}^{(i)} = \texttt{SelfAttention}(\mathbf{X}^{(i-1)}; \theta_s)$$
$$\mathbf{X}^{(i)} = \texttt{FFN}(\mathbf{H}^{(i)}; \theta_f) \tag{4}$$

The contextual presentations $\mathbf{X}^{(N)}$ of the last layer is used to recover the masked tokens $\bar{\boldsymbol{x}}$. Thus the training loss of inner-sequence masked language modeling can be formalized as:

$$\mathcal{L}_{\text{IS-MLM}}(\boldsymbol{x}) = -\log P(\bar{\boldsymbol{x}}|\hat{\boldsymbol{x}}; \theta_s, \theta_f) \tag{5}$$

**CS-MLM: Cross-Sequence Masked Language Modeling** In order to fully train the cross-attention module that plays a primary role in semantic mapping between sentences in cross-lingual generation tasks (e.g., machine translation), we simulate a decoder by reusing the self-attention and FFN modules to cooperate with the cross-attention module. As shown in Figure 1, the green lines depict the forward process of the CS-MLM task in each layer. Specifically, we first extract the contextual representation of $\hat{\boldsymbol{y}}$ via the $\texttt{SelfAttention}$ module[3]; then a cross-attention module is employed to model an interactive representation of $(\hat{\boldsymbol{x}}, \hat{\boldsymbol{y}})$:

$$\mathbf{S}^{(i)} = \texttt{SelfAttention}(\mathbf{Y}^{(i-1)}; \theta_s)$$
$$\mathbf{Z}^{(i)} = \texttt{CrossAttention}(\mathbf{S}^{(i)}, \mathbf{X}^{(N)}; \theta_c) \tag{6}$$
$$\mathbf{Y}^{(i)} = \texttt{FFN}(\mathbf{Z}^{(i)}; \theta_f)$$

---

[3]A triangular attention mask matrix is used to only attend to the preceding tokens of each masked token.

Finally, $\mathbf{Y}^{(N)}$, considering both the context of the semantic-related sequence $\hat{\boldsymbol{x}}$ and its left segments $\hat{\boldsymbol{y}}_{<t}$, is used to predict the masked tokens $\bar{\boldsymbol{y}}$:

$$\mathcal{L}_{\mathrm{CS-MLM}}(\boldsymbol{x}, \boldsymbol{y}) = -\log P(\bar{\boldsymbol{y}}|\hat{\boldsymbol{x}}, \hat{\boldsymbol{y}}_{<t}; \theta_s, \theta_c, \theta_f) \tag{7}$$

Note that when optimizing the CS-MLM objective, we detach $\mathbf{X}^{(N)}$ in Eq. (6) from the computation graph (i.e., stop the gradients back-propagation from "virtual" decoder to "virtual" encoder) to let the two objectives optimized in isolation. It also speeds up and stabilizes the training of this "virtual" decoder model, since very deep encoder-decoders are typically hard to train.

To conclude, the total MLM loss for a training instance $(\boldsymbol{x}, \boldsymbol{y})$, by exchanging the $\hat{\boldsymbol{x}}$ and $\hat{\boldsymbol{y}}$ in Eq. (5) and Eq. (7) [4], can be further formalized as:

$$\begin{aligned}
\mathcal{L}_{\mathrm{MLM}}(\boldsymbol{x}, \boldsymbol{y}) &= \mathcal{L}_{\mathrm{IS-MLM}}(\boldsymbol{x}) + \mathcal{L}_{\mathrm{IS-MLM}}(\boldsymbol{y}) + \mathcal{L}_{\mathrm{CS-MLM}}(\boldsymbol{x}, \boldsymbol{y}) + \mathcal{L}_{\mathrm{CS-MLM}}(\boldsymbol{y}, \boldsymbol{x}) \\
&= -\Big(\log P(\bar{\boldsymbol{x}}|\hat{\boldsymbol{x}}) + \log P(\bar{\boldsymbol{y}}|\hat{\boldsymbol{y}}) + \log P(\bar{\boldsymbol{y}}|\hat{\boldsymbol{x}}, \hat{\boldsymbol{y}}_{<t}) + \log P(\bar{\boldsymbol{x}}|\hat{\boldsymbol{y}}, \hat{\boldsymbol{x}}_{<t})\Big)
\end{aligned} \tag{8}$$

Several monolingual pre-training models such as MASS (Song et al., 2019), BART (Lewis et al., 2019) and PALM (Bi et al., 2020) also present similar unidirectional language modeling tasks. Except that VECO focuses on the multilingual scenario, there are some major differences in terms of both model architecture and task design: 1) In terms of model architecture, the core difference is that we share the self-attention and FFN modules in the encoder and decoder. We find that such parameter sharing can enhance the semantic mapping between different languages not only at the embedding-level (e.g, shared BPE vocabulary), but also at the module-level. Moreover, it also acts as a form of regularization that stabilizes the training and helps with generalization (Xia et al., 2019; Lan et al., 2019). 2) In terms of task design, we differ in several ways. The proposed IS-MLM task forces the model to bidirectionally comprehend the source input (good for NLU), which is a shortage of mBART. Meanwhile, CS-MLM predicts the masked words other than generating the next word (adopted by MASS and mBART), thus keeping in line with IS-MLM towards a more consistent optimization direction (predicting masked words) on the shared parameters.

In total, the core contribution of this article is the exquisite cooperation of parameters sharing and pre-training strategy, making VECO flexibly initialize any downstream framework.

## 2.3 Fine-tuning on Downstream NLU and NLG Tasks

When fine-tuning on various downstream tasks, one advantage of VECO is its flexibility for initializing both the encoder-only Transformer and encoder-decoder Transformer.

**VECO for Cross-lingual Natural Language Understanding**  Since the mainstream framework for NLU is an encoder-only Transformer, we only keep the self-attention and FFN modules while throwing out the cross-attention module in each layer (Figure 1(b)). Note that the cross-attention modules only occupy less than 20% of the total parameters, which is smaller than the discarded generator of ELECTRA (Clark et al., 2020b).

**VECO for Cross-lingual Natural Language Generation**  Considering the most popular backbone network for cross-lingual generation tasks like machine translation is the standard Transformer encoder-decoder model, we reorganize the VECO modules to act like that. As shown in Figure 1(c), the self-attention and FFN modules constitute the Transformer encoder for contextual modeling, and the three modules work together to act as a decoder for both inner-sequence and cross-sequence contextual modeling. Due to the training difficulty and inference speed of the deep network, we can choose a subset (e.g., 6 layers) of all layers to assemble the decoder. More in-depth analysis can be found in Table 4 and Section 5.2.

## 3 Pre-training Setup

**Model Configuration**  We pre-train a 24-layer model with 1024 embedding/hidden size and 4096 feed-forward size ($\sim$ 662M parameters). We do not use language embeddings to allow our model to

---

[4] Flipping the monolingual sentence order can create "harder" example pairs, thus pushing the model toward a stronger ability of language modeling and understanding.

better deal with downstream tasks of unseen languages. We adopt the same 250K vocabulary that is also used by XLM-R (Conneau et al., 2019) and mBART (Liu et al., 2020b).

**Pre-Training Datasets**   For monolingual training datasets, we reconstruct Common-Crawl Corpus used in XLM-R (Conneau et al., 2019). We extract 1.36TB data in 50 languages, which contains 6.5G sentences and 0.4G documents. We up/down-sample the monolingual text like XLM from each language with a smoothing parameter $\alpha = 0.5$. For bilingual data, we collect from the OPUS website[5] like previous works (Lample & Conneau, 2019; Chi et al., 2020b). There are 6.4G parallel sentences, covering 879 language pairs across 50 languages. More details about the languages and statistics of our training corpus can be founded in Appendix A.

**Optimization Settings**   For each training iteration, we alternately sample a batch of adjacent segments from monolingual corpus and a batch of parallel sentences from bilingual datasets to conduct a pair of masked input $(\hat{x}, \hat{y})$. We firstly perform IS-MLM for both $\hat{x}$ and $\hat{y}$. Then, we reuse their hidden states from the last layer to perform CS-MLM. Meanwhile, we also adopt the translation masked language modeling (TLM) proposed in XLM (Lample & Conneau, 2019) when the inputs are parallel bilingual sentences. Thus the overall training objective is the sum of three MLM objectives. During training, the model parameters except for cross-attention are initialized by XLM-R. We first freeze the parameters of XLM-R and only update the cross-attention parameters for faster convergence. Then, we jointly train the whole model. We pre-train our model with mixed-precision training using 64 Nvidia Telsa V100 32GB GPUs. More hyperparameters can be founded in Table 7.

## 4 EXPERIMENTS ON LANGUAGE UNDERSTANDING

### 4.1 EXPERIMENTAL SETUP

**Downstream Tasks**   We conduct NLU evaluations on XTREME (Hu et al., 2020), a representative massively multilingual multi-task benchmark. It consists of various NLU tasks over 40 languages. XTREME tasks can be classified into four different categories: (1) sentence-pair classification: XNLI (Conneau et al., 2018), PAWS-X (Yang et al., 2019); (2) structured prediction: UD-POS (Nivre et al., 2018), Wikiann NER (Pan et al., 2017); (3) question answering: XQuAD (Artetxe et al., 2020), MLQA (Lewis et al., 2020), TyDiQA (Clark et al., 2020a); (4) sentence retrieval: BUCC 2018 (Zweigenbaum et al., 2018), Tatoeba (Artetxe & Schwenk, 2019). Tasks in the first three categories only provide golden training corpus in English and dev/test set in different target languages. For the two zero-shot sentence retrieval tasks, no training datasets are provided. We refer the reader to Hu et al. (2020) for additional details about the datasets.

**Fine-tuning Setting**   Following previous works (Conneau et al., 2019; Hu et al., 2020), we consider two typical fine-tuning settings: (1) *Cross-lingual Transfer* which fine-tunes the pre-trained model only using English golden data and directly performs inference on the test data of different target languages; (2) *Translate-Train-All* which first automatically translates the English golden data to the remaining target languages and fine-tunes a multilingual model on the concatenation of all data. We use the machine-translated data released by XTREME, except for two sequence-labeling tasks (POS, NER) since the golden token labels in the target language are unavailable. To have a fair comparison with the strong baseline XLM-R (Conneau et al., 2019) under the translate-train-all setting, we also show the results of XLM-R using the same fine-tuning hyperparameters as VECO.

### 4.2 EXPERIMENTAL RESULTS

The detailed test results of 9 tasks on the XTREME benchmark are shown in Table 1. It demonstrates that the proposed VECO outperforms previous cross-lingual models on most of the datasets. Compared to XLM-R, it averagely scores 5.0 and 2.8 points higher under the cross-lingual transfer and translation-train-all settings, respectively. It is worth noting that, VECO delivers a large improvement on *zero-shot* sentence retrieval tasks (BUCC, Tatoeba). This phenomenon reflects that our model has a strong cross-lingual modeling ability, thus it can better mine parallel sentences in a multilingual corpus. The reasons are two-folds. First is the introduction of more bilingual data

---

[5]http://opus.nlpl.eu/

| Model | Sentence Classification | | Structured Prediction | | Question Answering | | | Sentence Retrieval | | Avg. |
|---|---|---|---|---|---|---|---|---|---|---|
| | XNLI | PAWS-X | UD-POS | NER | XQuAD | MLQA | TyDiQA | BUCC | Tatoeba | |
| | Acc | Acc | F1 | F1 | F1/EM | F1/EM | F1/EM | F1 | Acc | |
| *Cross-lingual Transfer: Fine-tune model on English training set and test on all languages* | | | | | | | | | | |
| MMTE[†] | 67.4 | 81.3 | 73.5 | 58.3 | 64.4/46.2 | 60.3/41.4 | 58.1/43.8 | 59.8 | 37.9 | 59.5 |
| mBERT[†] | 65.4 | 81.9 | 70.3 | 62.2 | 64.5/49.4 | 61.4/44.2 | 59.7/43.0 | 56.7 | 38.7 | 59.6 |
| XLM[†] | 69.1 | 80.9 | 70.1 | 61.2 | 59.8/44.3 | 48.5/32.6 | 43.6/29.1 | 56.8 | 32.6 | 55.5 |
| XLM-R[†] | 79.2 | 86.4 | 72.6 | 65.4 | 76.6/60.8 | 71.6/**53.2** | 65.1/45.0 | 66.0 | 57.3 | 68.1 |
| VECO | **79.9** | **88.7** | **75.1** | **65.7** | **77.3/61.8** | **71.7/53.2** | **67.6/49.1** | **85.0** | **75.1** | **73.1** |
| *Translate-Train-All: Fine-tune model on English training data and translated data of the target language* | | | | | | | | | | |
| XLM-R[‡] | 82.6 | 90.4 | - | - | **80.2**/65.9 | 72.8/54.3 | 66.5/47.7 | - | - | - |
| XLM-R[*] | 82.8 | 90.2 | 72.6 | 65.4 | 80.0/65.8 | 73.0/54.3 | 74.5/58.3 | 80.2 | 75.2 | 74.4 |
| VECO | **83.0** | **91.1** | **75.1** | **65.7** | 79.9/**66.3** | **73.1/54.9** | **75.0/58.9** | **89.3** | **86.9** | **77.2** |

Table 1: XTREME results on each dataset (as of Oct 02, 2020). Averaged results on the four categories can be found at leaderboard: `https://sites.research.google/xtreme`. "†" and "‡" indicates results from Hu et al. (2020) and Fang et al. (2020), respectively. "*" indicates the results obtained by our implementation. The detailed results for each language are in Appendix C.

| Models | Datasets | | Tasks | | | Languages | | | | | | | | | Avg. |
|---|---|---|---|---|---|---|---|---|---|---|---|---|---|---|---|
| | Mono. | Bili. | IS-MLM | CS-MLM | TLM | en | de | es | fr | ru | tr | th | vi | zh | |
| XLM$_{\text{SMALL}}$ | ✓ | | ✓ | | | 76.6 | 58.0 | 63.5 | 62.0 | 56.1 | 54.8 | 54.4 | 57.6 | 55.5 | 59.8 |
| VECO$_{\text{SMALL}}$ | ✓ | | ✓ | ✓ | | 77.1 | 60.9 | 64.7 | 62.5 | 56.1 | 55.3 | 54.7 | 57.5 | 56.6 | 60.6 |
| XLM$_{\text{SMALL}}$ | ✓ | ✓ | ✓ | | ✓ | 78.3 | 55.1 | 72.1 | 72.5 | 68.2 | 51.1 | 48.8 | 51.6 | 52.9 | 61.2 |
| VECO$_{\text{SMALL}}$ | ✓ | ✓ | ✓ | ✓ | ✓ | 80.0 | 66.4 | 75.6 | 75.6 | 70.9 | 58.7 | 56.1 | 63.3 | 62.9 | 67.7 |

Table 2: XNLI accuracy scores for each language under the cross-lingual transfer setting. Both XLM and VECO are small-sized models trained from scratch on the same monolingual (Mono.) and bilingual (Bili.) corpus using the same hyperparameters. For this set of experiments, we only use a subset of the full training corpus and report the results of languages appeared in them.

during pre-training, which is a direct and effective way to enhance the cross-lingual ability of the model. Second is the mutual improvement between two pre-training tasks and the superiority of model design.

To analyze whether bilingual data plays a leading role in improving performance, we conduct a set of more fair experiments. We train small-sized XLM and VECO models from scratch using a subset of full training data and hyperparameters (see Appendix A for details). Table 2 shows the results of XNLI, the most widely used cross-lingual evaluation dataset in the XTREME leaderboard. We observe that, when using monolingual corpus only, VECO can outperform XLM by 0.8 points. It suggests that our models can benefit from adjacent sentences used by the CS-MLM task to be equipped with a stronger ability of contextual modeling. Moreover, when trained on both the monolingual and bilingual corpus, VECO can achieve a larger improvement compared to XLM. It reveals that VECO can better utilize the bilingual corpus, compared to only-optimized translation language modeling (TLM) in XLM.

## 5 EXPERIMENTS ON LANGUAGE GENERATION

### 5.1 EXPERIMENTAL SETUP

**Datasets** We choose machine translation (MT) task, a typical cross-lingual generation scenario. In order to illustrate the generality of our approach and have a fair comparison with the most recent state-of-the-art Transformer works (Liu et al., 2020a), we choose two most widely used datasets: WMT14 English→German (En-De) and English→French (En-Fr) translation. WMT14 En-De is a medium-resource dataset which provides 4.5M pairs for training and validation. We adopt standard newstest2014 as the test set. WMT14 En-Fr is a high-resource dataset which contains 36M pairs of parallel sentences. We use newstest2012+newstest2013 for validation and newstest2016 for test. We measure case-insensitive tokenized BLEU with `multi-bleu.perl` to have a fair compari-

| Model | #Layers | | WMT14 En-Fr | | WMT14 En-De | |
|---|---|---|---|---|---|---|
| | Enc | Dec | BLEU | sacreBLEU | BLEU | sacreBLEU |
| *Randomly Initialized* | | | | | | |
| Transformer-Big (Vaswani et al., 2017) | 6 | 6 | 41.0 | - | 28.4 | - |
| Transformer-xBig (Our impl.) | 24 | 6 | 42.8 | 40.3 | 28.6 | 27.7 |
| Deep-Transfromer (Liu et al., 2020a) | 60 | 12 | 43.8 | 41.8 | 30.1 | 29.5 |
| *Cross-lingual Models Initialized* | | | | | | |
| XLM (Lample & Conneau, 2019) | 6 | 6 | - | - | 28.9 | - |
| XLM-R (Conneau et al., 2019) | 24 | 6 | 43.7 | 41.1 | 30.8 | 29.9 |
| mBART (Liu et al., 2020b) | 12 | 12 | 43.2 | 41.0 | 30.0 | 29.1 |
| VECO | 24 | 6 | **44.4** | **42.0** | **31.5** | **30.5** |

Table 3: Results on WMT14 En-Fr and WMT14 En-De Machine Translation.

son with previous Transformer variants and de-tokenized `SacreBLEU`[6] to avoid the influence of different tokenization and normalization between models (Post, 2018).

**Fine-tuning Setting** In theory, VECO can initialize a 24-layer encoder and 24-layer decoder Transformer model. However, since VECO follows the post-layernorm design of vanilla Transformer (Vaswani et al., 2017) like XLM-R, we find it is hard to train such a deep post-layernorm Transformer variant without careful parameter searching. This phenomenon is consistent with the findings in recent researches about deep transformer (Liu et al., 2020a; Bachlechner et al., 2020). Since these works also show that deeper encoders are more worthwhile than deeper decoders, thus the main results of VECO and XLM-R are based on the most normal 6-layer decoder with the full 24-layer encoder. The batch size is 64k and 256k for En-De and En-Fr respectively. The total training updates are set to 100k. The learning rate is 1e-4/2e-4, with linear warmup over the first 16k steps and linear decay. We run En-De and En-Fr MT experiments on 16 and 32 V100 GPUs respectively. We average the last 10 checkpoints and use beam search with a beam size of 5.

**Baselines** We consider two types of Transformer baselines: randomly initialized and cross-lingual models initialized. For random initialization, we take the original Transformer-big and the state-of-the-art Deep Transformer (Liu et al., 2020a) into consideration. Besides, we also reproduce a Transformer baseline which adopts the same fine-tuning hyperparameters as VECO but with random initialization. For cross-lingual encoder-decoder models, we include mBART, which shows impressive results on MT. We also conduct the WMT experiments for XLM-R, following the totally same fine-tuning settings as VECO. Note that the layer settings are not totally same, due to the distinct characteristic of pre-trained models: (1) The layer of pre-trained encoder-decoder models decide the layer of MT experiments. Thus the layer of mBART is set the same as pre-training (ie, 12-layer encoder and 12-layer decoder); (2) The layer of pre-trained encoder models only decides the layer of the MT encoder, while the decoder layer can be any value. Thus XLM-R initialized model is fixed as a 24-layer encoder during fine-tuning. In order to minimize the layer gap between XLM-R and mBART initialized MT model, we choose the most common 6-layer decoder in Table 3 and 3-layer decoder in Table 4. To have a fair comparison to XLM-R, VECO also adopts the same layer settings. We also reproduce a same-sized randomly initialized Transformer-xBig model. In conlusion, we tried our best to use the same layer setting among most of the models if possible (Transformer-xBig, XLM-R, and VECO).

## 5.2 EXPERIMENTAL RESULTS

Table 3 shows the comparison between VECO and baselines. Compared to existing cross-lingual models, VECO can consistently outperform the best models about 1 BLEU. We can also observe that VECO can largely outperform randomly initialized same-sized Transformer-xBig model by 2 BLEU. And it even beats the state-of-the-art Deep-Transformer with a 60-layer encoder and 12-layer decoder on both datasets.

---

[6]Hash: BLEU+case.mixed+lang.en-{de,fr}+numrefs.1+smooth.exp+test.wmt14/full+tok.13a+version.1.4.9

| Method | #Layers | | WMT14 En-De | |
|---|---|---|---|---|
| | Enc | Dec | BLEU | sacreBLEU |
| *Randomly Initialize* | 24 | 3 | 28.5 | 27.6 |
| | 24 | 6 | 28.6 | 27.7 |
| *Randomly Initialize + More Bilingual Data\** | | | | |
| | 24 | 6 | 30.4 | 29.4 |
| *VECO Initialize* | 24 | First-3 | 30.8 | 29.8 |
| | 24 | Last-3 | 31.2 | 30.3 |
| | 24 | First-6 | 31.1 | 30.1 |
| | 24 | Last-6 | **31.5** | **30.5** |

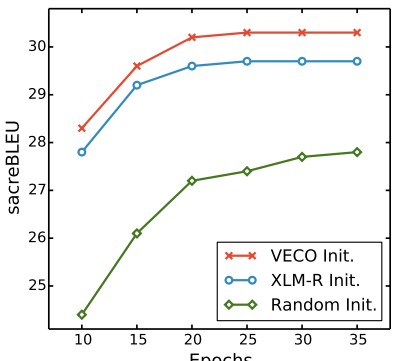

Table 4: BLEU scores (left) and learning curves (right) of different initialization methods.

Table 4 (left) contrasts two ways initializing a Transformer with $n(< 24)$ decoder layers via selecting: (1) the first $n$ layers; (2) the last $n$ layers from a 24-layer pre-trained VECO model. We consider $n = \{3, 6\}$ to conduct experiments. The predominant method adopted in this paper, the straightforward strategy of selecting the last $n$ layers, exhibits better performance. It is possibly because the last several layers play a more important role in making predictions over the whole vocabulary. Moreover, we can find that there is 0.2~0.3 BLEU improvement when increasing the decoder layers from 3 to 6.[7] Regardless of the initialization method, the VECO-initialized model can gain consistent 1~2 BLEU improvement over the randomly-initialized model.

Moreover, Table 4 (right) displays the sacreBLEU scores of same-sized (24-layer encoder and 6-layer decoder) models during training. We find that VECO-initialized model can get a surprising more than 28 sacreBLEU score just after 10 epochs, which is better than the final score of the randomly initialized model at 35 epochs. It reveals that VECO can provide a fairly good initialization for the machine translation model, which can converge quickly and further boost the results.

To investigate whether the exciting improvement in MT mainly comes from 1) the use of parallel corpus during pre-training or 2) the superiority of the designed model and pre-training tasks, we also conduct a more comparable experiment. We first train an out-of-domain Transformer-xBig model using the whole En-De parallel data ($\sim$ 68M) used in VECO pre-training, and then continue to train the model on the in-domain WMT14 En-De training dataset. Results are shown in Table 4 (left) marked with *. Under this set of a totally fair comparison, VECO still maintains a lead of 1.1 BLEU score. This directly confirms that the improvement on MT is not only due to the use of bilingual data. More importantly, reasonable pre-training tasks and model design ensure *better* use of bilingual and large-scale unlabeled multilingual corpus.

# 6    RELATED WORK

**Encoder-only Cross-lingual Pre-training**  mBERT (Devlin et al., 2019) is the first work to pre-train a Transformer encoder over multiple languages. There have been several extensions that use the same encoder-only backbone as mBERT, with the main difference is the introduction of more training corpus and pre-training tasks. XLM (Lample & Conneau, 2019) utilizes both monolingual and bilingual corpus to perform mask language modeling. XLM-R (Conneau et al., 2019) extends to be built on RoBERTa (Liu et al., 2019) using larger monolingual training data. Unicoder (Huang et al., 2019), ALM (Yang et al., 2020), and InfoXLM (Chi et al., 2020b) propose new pre-training tasks to better utilize the bilingual data. These works deliver impressive performance on cross-lingual understanding tasks, while only marginal improvement has been gained on cross-lingual generation tasks like machine translation, especially on high-resource languages.

**Encoder-Decoder Cross-lingual Pre-training**  BART (Lewis et al., 2019) pre-trains denoising auto-encoder built with a standard Transformer-based encoder-decoder architecture.  And

---

[7]However, we observe that only marginal improvement can be gained when further increasing the decoder layers to 12 or 24 in our primary experiments, which is also in line with the findings in Liu et al. (2020a).

mBART (Liu et al., 2020b) extends it to the multilingual setting, demonstrating significant gains in low/medium-resource machine translation, but with a decrease in high resource languages. Chi et al. (2020a) first trains an encoder via MLM and then frozen the encoder to train the decoder only via two generative tasks. A similar approach is also proposed in Liang et al. (2020) that extends Unicoder (Huang et al., 2019) to generation tasks, with the main difference exists in the joint training of encoder-decoder. All these cross-lingual models emphasize training a dedicated model for NLG, thus these didn't improve or even hurt the NLU capabilities of the encoder. Or they require more computation and memory to match the performance of the encoder-only models when using comparable training resources. For example, BART, when fine-tuning on NLU tasks, the same input is fed into the encoder and decoder, and the final output from the decoder is used. Thus it would cost more memory due to extra cross-attention modules (roughly 10%∼20% more parameters) than encoder-only RoBERTa model, but BART still can't performs better than RoBERTa.

**Unified Language Representation for NLU and NLG** BART (Lewis et al., 2019) and UNILM (Dong et al., 2019) also endeavor to build a unified model for NLU and NLG tasks, with the core idea that allows the model to see both unidirectional and bidirectional context. UNILM pre-trains a Transformer encoder with an ensemble of attention masks, while BART pre-trains a Transformer encoder-decoder model with arbitrary noising functions. Our work differs from them in several ways. Firstly, they only focus on the monolingual (English) domain, while we dedicate to the more challenging multilingual domain. Secondly, since multiple languages cannot be completely mapped to the same space at every layer only via self-attention like UNILM, the cross-attention module is important to model the cross-lingual mapping. Thus we drop the way to pre-train an encoder-only model under the multilingual scenario. Thirdly, previous work (Xia et al., 2019) have shown that sharing the encoder and decoder of the Transformer can strengthen the semantic correlation between languages and regularize the encoder-decoder with high capacity on machine translation. Due to the above reasons, VECO varies during training to train specific modules via a bidirectional task (IS-MLM) and unidirectional task (CS-MLM).

## 7 CONCLUSION

We present VECO, a variable cross-lingual pre-training model, targeted at initializing both NLU preferred encoder-only and NLG specialized encoder-decoder Transformer. We analyze three core modules in standard Transformer and propose two masked language modeling tasks to train the reasonable combinations of them. The two tasks jointly optimize for inner-sequence understanding and cross-sequence generation, enabling them to boost each other via strong regularization from module sharing. On this account, VECO achieves consistent improvements on various language understanding and generation tasks compared to existing cross-lingual encoder-only and encoder-decoder approaches, opening up new ways of thinking about pre-trained backbone architecture.

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

## A    PRE-TRAINING DETAILS

For monolingual data, following XLM-R (Conneau et al., 2019), we build a clean CommonCrawl Corpus using an open-source tool CCNet (Wenzek et al., 2019). Table 5 reports the language codes and statistics of datasets. There are 1.36TB monolingual data in 50 languages before up/down-sampling.

We collect bilingual corpus in 50 languages from the OPUS website[8], including MultiUN, UNPC, Bombay, EU-bookshop, OpenSubtitles2018, Tanzil, GlobalVoices, ParaCrawl, MultiParaCrawl, DGT, Tilde, Europarl, Wikipedia, ECB, TED2013, News-Commentary, Ubuntu, Books, UN, infopankki-v1, EUconst, and Bianet. In total, there are 1TB bilingual training corpus before preprocessing, covering 879 language pairs. Table 6 lists the statistics for each language pair.

We then apply subword tokenization directly on raw text data using Sentence Piece Model (Kudo & Richardson, 2018) without any additional preprocessing.

We use the whole corpus to train VECO, while using a subset ($\sim 1/4$) that contains 33 languages to train XLM$_{\text{SMALL}}$ and VECO$_{\text{SMALL}}$. The full set of pre-training hyperparameters for small-sized and large-sized VECO (default) are listed in Table 7. The comparisons between VECO and other cross-lingual models are shown in Table 8.

## B    NLU FINE-TUNING DETAILS

We consider two typical fine-tuning settings:

- *Cross-lingual Transfer*: For all tasks except sentence retrieve tasks (BUCC and Tatoeba), we select the model with the best average result over all the languages on the dev sets, by searching the learning rate over [1e-5,2e-5,3e-5], training epoch over [3,5,10], and batch size over [16,32,64]. As for sentence retrieve tasks without fine-tuning on any parallel sentences at all, we use the average word embeddings in the middle layer to extract the sentence representations.

- *Translate-Train-All*: We found that the large-sized training datasets can benefit from a smaller learning rate. Therefore, for translate-train-all, we select the model on the dev sets, by searching the learning rate over [3e-6,5e-6,1e-5]. For XNLI, PAWS-X, XQuAD and MLQA, we fine-tune our model directly on the translation data provided by the official XTREME repo[9]. Following the participants (FILTER and Anonymous1) on the XTREME leaderboard, we use several methods to further improve the scores. For TyDiQA, we start fine-tuning based on the best XQuAD translate-train-all model. For the sentence retrieve tasks, BUCC and Tatoeba, we use the averaged representation in the middle layer of the best XNLI model. Note that we only use these tricks under the translate-train-all setting, wishing to have a possible fair comparison on the XTREME leaderboard. However, we still use less fine-tuned data compared to concurrent works FILTER and Anonymous1. Specifically, FILTER, a model-agnostic fine-tune method, uses more translation data on POS and NER tasks. Anonymous1 uses other labeled data outside of XTREME.

---

[8]http://opus.nlpl.eu/
[9]https://github.com/google-research/xtreme

| Language | #Document(M) | #Sentence(M) | Size(GB) |
|---|---|---|---|
| af | 0.023 | 0.522 | 0.107 |
| ar | 2.823 | 42.659 | 11.786 |
| bg | 0.919 | 14.743 | 5.217 |
| bn | 0.750 | 9.217 | 4.264 |
| cs | 3.980 | 55.754 | 9.668 |
| de | 21.410 | 310.942 | 66.333 |
| el | 1.740 | 24.334 | 9.737 |
| en | 130.087 | 2,215.534 | 479.099 |
| es | 17.569 | 267.764 | 58.774 |
| et | 0.347 | 5.252 | 0.877 |
| eu | 0.342 | 5.216 | 0.613 |
| fr | 15.819 | 267.888 | 58.023 |
| fa | 2.506 | 43.570 | 13.831 |
| fi | 1.530 | 23.790 | 3.940 |
| fy | 0.027 | 0.537 | 0.054 |
| gu | 0.039 | 0.519 | 0.228 |
| gd | 0.009 | 0.126 | 0.020 |
| he | 0.755 | 12.338 | 3.073 |
| hi | 0.536 | 7.303 | 3.762 |
| hu | 1.816 | 29.962 | 6.421 |
| id | 3.417 | 60.908 | 11.528 |
| it | 9.336 | 133.006 | 30.854 |
| ja | 27.967 | 588.926 | 71.785 |
| jv | 0.002 | 0.138 | 0.030 |
| ka | 0.141 | 1.756 | 0.766 |
| kk | 0.061 | 1.545 | 0.448 |
| ko | 11.609 | 227.396 | 27.837 |
| lt | 0.552 | 7.996 | 1.480 |
| lv | 0.281 | 4.159 | 0.798 |
| ms | 0.334 | 3.762 | 0.455 |
| ml | 0.162 | 2.615 | 1.025 |
| my | 0.045 | 0.893 | 0.306 |
| mr | 0.059 | 0.708 | 0.365 |
| pl | 6.642 | 93.760 | 19.082 |
| pt | 8.623 | 128.107 | 25.612 |
| ne | 0.080 | 0.829 | 0.429 |
| nl | 6.513 | 85.997 | 16.648 |
| ru | 35.887 | 580.291 | 203.105 |
| ro | 1.944 | 31.929 | 7.056 |
| si | 0.132 | 2.927 | 0.902 |
| sw | 0.057 | 0.945 | 0.179 |
| ta | 0.876 | 20.376 | 6.422 |
| te | 0.288 | 4.995 | 1.721 |
| tr | 18.547 | 291.081 | 40.321 |
| th | 6.278 | 117.826 | 27.941 |
| tl | 0.166 | 5.611 | 0.679 |
| vi | 12.183 | 234.071 | 37.919 |
| ur | 0.460 | 7.509 | 2.003 |
| yo | 0.0002 | 0.003 | 0.0005 |
| zh | 27.067 | 497.408 | 87.005 |
| **Total** | 382.735 | 6,475.444 | 1,360.526 |

Table 5: The statistics of monolingual pre-training corpus.

| Pair | #Sent(K) | Pair | #Sent(K) | Pair | #Sent(K) | Pair | #Sent(K) | Pair | #Sent(K) | Pair | #Sent(K) | Pair | #Sent(K) | Pair | #Sent(K) | Pair | #Sent(K) |
|---|---|---|---|---|---|---|---|---|---|---|---|---|---|---|---|---|---|
| af-ar | 12.34 | bg-my | 0.08 | de-he | 12751.69 | en-tr | 46584.82 | eu-zh | 19.76 | fy-vi | 34.95 | id-pt | 6825.29 | ko-sw | 6.74 | pl-es | 46863.47 |
| af-bg | 18.19 | bg-ne | 0.01 | de-hi | 106.11 | en-ur | 781.60 | fa-fi | 4485.62 | gd-es | 21.62 | id-ro | 7944.59 | ko-ta | 13.74 | pl-pt | 72437.93 |
| af-bn | 1.19 | bg-nl | 30757.50 | de-hu | 24409.40 | en-vi | 3563.39 | fa-fr | 4507.06 | gd-it | 13.26 | id-ru | 5039.44 | ko-te | 0.93 | pl-ru | 19170.23 |
| af-cs | 17.93 | bg-pl | 33043.03 | de-it | 35936.62 | en-yo | 0.13 | fa-he | 4944.80 | gd-pl | 12.29 | id-si | 366.00 | ko-th | 230.84 | pl-sw | 1424.02 |
| af-de | 19.28 | bg-pt | 30058.54 | de-ja | 1472.72 | en-zh | 28952.02 | fa-hi | 186.23 | gd-pt | 18.90 | id-sw | 30.56 | ko-tl | 1.21 | pl-tl | 1039.37 |
| af-el | 29.83 | bg-ro | 38925.52 | de-ka | 123.12 | es-et | 18090.74 | fa-hu | 5201.51 | gd-tr | 14.12 | id-te | 35.37 | ko-tr | 1246.58 | pl-tr | 32470.18 |
| af-en | 44.70 | bg-ru | 17423.43 | de-kk | 3.72 | es-eu | 793.59 | fa-id | 3220.00 | he-hi | 57.85 | id-th | 35.37 | ko-ur | 57.21 | pl-ur | 391.99 |
| af-es | 34.31 | bg-si | 460.50 | de-ko | 776.89 | es-fa | 5696.70 | fa-it | 4243.56 | he-hu | 23959.87 | id-tl | 7.80 | ko-vi | 345.79 | pl-vi | 3790.71 |
| af-et | 6.34 | bg-sw | 10.80 | de-lt | 9134.99 | es-fi | 34222.07 | fa-ja | 1072.14 | he-id | 6362.29 | id-tr | 8017.99 | ko-zh | 56.43 | pt-ro | 33802.95 |
| af-fa | 3.07 | bg-ta | 27.14 | de-lv | 8532.06 | es-fr | 96233.21 | fa-kk | 1.01 | he-it | 19908.66 | id-ur | 172.71 | lt-lv | 6546.76 | pt-si | 450.40 |
| af-fi | 10.25 | bg-te | 17.14 | de-ml | 294.16 | es-he | 27060.49 | fa-ko | 627.97 | he-ja | 1683.29 | id-vi | 2081.70 | lt-ml | 66.40 | pt-sw | 13.06 |
| af-fr | 18.56 | bg-th | 2733.84 | de-my | 0.68 | es-hi | 85.35 | fa-lt | 615.78 | he-ka | 149.06 | id-zh | 356.46 | lt-ms | 393.89 | pt-ta | 26.37 |
| af-fy | 36.94 | bg-tl | 6.69 | de-ne | 0.28 | es-hu | 43947.78 | fa-lv | 228.40 | he-kk | 2.38 | it-ka | 106.70 | lt-nl | 7497.18 | pt-te | 19.32 |
| af-he | 14.53 | bg-tr | 31179.35 | de-nl | 34909.49 | es-id | 8015.69 | fa-ml | 308.49 | he-ko | 1094.72 | it-kk | 2.54 | lt-pl | 9965.36 | pt-th | 2561.09 |
| af-hi | 1.15 | bg-ur | 71.60 | de-ro | 24261.82 | es-it | 49423.51 | fa-ms | 1072.22 | he-lt | 1220.91 | it-ko | 1125.97 | lt-pt | 7663.84 | pt-tl | 10.35 |
| af-hu | 16.32 | bg-vi | 2855.13 | de-ru | 10904.25 | es-ja | 1929.41 | fa-my | 0.06 | he-lv | 461.81 | it-lt | 7359.92 | lt-ro | 5786.22 | pt-tr | 27428.79 |
| af-id | 4.56 | bg-zh | 746.27 | de-si | 324.86 | es-ka | 181.19 | fa-ne | 0.01 | he-ml | 250.07 | it-ml | 235.96 | lt-ru | 950.02 | pt-ur | 73.57 |
| af-it | 15.01 | bn-cs | 340.51 | de-te | 12.81 | es-kk | 2.48 | fa-nl | 5010.64 | he-ms | 1455.61 | it-my | 0.36 | lt-si | 106.53 | pt-vi | 2963.83 |
| af-ja | 1.98 | bn-de | 346.51 | de-th | 1695.53 | es-ko | 1229.50 | fa-pt | 4998.09 | he-my | 0.05 | it-nl | 37644.29 | lt-sw | 0.02 | pt-yo | 0.05 |
| af-lt | 0.65 | bn-el | 340.94 | de-tl | 12.91 | es-lt | 7702.99 | fa-ro | 5714.73 | he-nl | 22186.61 | it-pt | 35301.98 | lt-ta | 13.04 | pt-zh | 846.44 |
| af-lv | 1.08 | bn-en | 752.08 | de-tr | 17579.53 | es-lv | 6703.10 | fa-ru | 4205.20 | he-pl | 24962.23 | it-ro | 32153.38 | lt-te | 9.71 | ro-ru | 19568.56 |
| af-ml | 2.18 | bn-es | 480.35 | de-ur | 218.89 | es-ml | 339.71 | fa-si | 292.78 | he-ro | 26370.15 | it-ru | 17669.12 | lt-th | 263.89 | ro-si | 504.24 |
| af-ms | 1.31 | bn-et | 252.68 | de-vi | 2284.70 | es-ms | 1731.36 | fa-sw | 69.51 | he-ru | 14873.77 | it-sw | 15.77 | lt-tl | 1.36 | ro-sw | 10.72 |
| af-nl | 22.61 | bn-eu | 42.42 | de-zh | 587.96 | es-my | 2.50 | fa-ta | 83.30 | he-si | 435.87 | it-th | 2447.55 | lt-tr | 1377.40 | ro-ta | 33.50 |
| af-pl | 1096.89 | bn-fa | 391.89 | el-en | 55078.46 | es-ne | 2.87 | fa-te | 10.11 | he-sw | 0.06 | it-tl | 13.30 | lt-vi | 486.84 | ro-te | 24.44 |
| af-pt | 22.68 | bn-fi | 279.35 | el-es | 46876.21 | es-nl | 46908.79 | fa-th | 1201.04 | he-ta | 23.99 | it-tr | 25770.29 | lt-zh | 40.65 | ro-th | 2874.73 |
| af-ro | 32.19 | bn-fr | 373.13 | el-et | 16463.57 | es-pt | 47542.26 | fa-tl | 7.02 | he-te | 18.65 | it-ur | 69.89 | lv-ml | 23.32 | ro-tl | 8.61 |
| af-ru | 15.41 | bn-he | 302.62 | el-eu | 673.93 | es-ro | 48229.60 | fa-ur | 568.00 | he-th | 2666.00 | it-vi | 2542.41 | lv-nl | 6622.81 | ro-tr | 36549.61 |
| af-si | 0.98 | bn-hi | 38.68 | el-fa | 5137.52 | es-ru | 55569.05 | fa-vi | 1514.04 | he-tr | 25179.32 | it-yo | 0.10 | lv-pl | 9460.93 | ro-ur | 73.55 |
| af-ta | 1.13 | bn-hu | 321.36 | el-fi | 28885.65 | es-si | 512.22 | fa-zh | 372.10 | he-ur | 20.57 | ja-ka | 35.37 | lv-pt | 6672.14 | ro-vi | 3207.73 |
| af-th | 2.08 | bn-id | 360.65 | el-fr | 38560.84 | es-sw | 41.33 | fi-he | 17820.49 | he-vi | 2813.73 | ja-ko | 308.30 | lv-ru | 435.73 | ro-zh | 947.91 |
| af-tr | 24.22 | bn-it | 301.31 | el-hi | 62.26 | es-ta | 31.19 | fi-hi | 55.60 | he-zh | 563.24 | ja-lt | 281.74 | lv-si | 34.42 | ru-si | 340.11 |
| af-vi | 3.30 | bn-ja | 142.19 | el-hu | 34559.75 | es-te | 21.76 | fi-hu | 27350.30 | hi-hu | 60.05 | ja-lv | 99.97 | lv-sw | 0.01 | ru-sw | 84.77 |
| ar-bg | 23090.32 | bn-ka | 8.68 | el-id | 7098.25 | es-th | 2976.49 | fi-id | 6306.36 | hi-id | 85.85 | ja-ml | 79.78 | lv-te | 4.01 | ru-ta | 61.50 |
| ar-bn | 378.28 | bn-ko | 93.92 | el-it | 34337.63 | es-tl | 13.55 | fi-it | 26756.85 | hi-it | 60.12 | ja-ms | 489.33 | lv-tr | 515.30 | ru-te | 10.80 |
| ar-cs | 24147.25 | bn-lt | 96.24 | el-ja | 1740.08 | es-tr | 39805.02 | fi-ja | 1599.82 | hi-ja | 46.14 | ja-nl | 3295.60 | lv-ur | 1.08 | ru-tl | 13.43 |
| ar-de | 12733.65 | bn-lv | 41.21 | el-ka | 167.39 | es-ur | 79.44 | fi-ka | 148.42 | hi-ka | 0.80 | ja-pl | 3295.60 | lv-vi | 209.40 | ru-tr | 19317.60 |
| ar-el | 22486.60 | bn-ml | 93.14 | el-kk | 2.33 | es-vi | 3215.16 | fi-kk | 3.41 | hi-ko | 33.66 | ja-ro | 1843.14 | lv-zh | 14.71 | ru-ur | 417.23 |
| ar-en | 60392.55 | bn-ms | 203.84 | el-ko | 1130.94 | es-yo | 0.12 | fi-ko | 859.31 | hi-lt | 23.67 | ja-si | 162.96 | ml-ms | 101.75 | ru-vi | 2289.72 |
| ar-es | 57561.29 | bn-my | 0.78 | el-lt | 7400.42 | es-zh | 28688.60 | fi-lt | 12.61 | hi-lv | 12.61 | ja-ta | 18.92 | ml-nl | 268.10 | ru-yo | 0.10 |
| ar-et | 9738.71 | bn-ne | 0.78 | el-lv | 6549.40 | et-fa | 3085.41 | fi-lv | 6732.38 | hi-ml | 30.28 | ja-th | 632.26 | ml-pt | 280.62 | ru-zh | 28138.59 |
| ar-eu | 578.30 | bn-nl | 331.34 | el-ml | 302.85 | et-fi | 15969.08 | fi-ml | 232.48 | hi-ms | 40.38 | ja-tr | 1896.56 | ml-ro | 325.97 | si-ta | 6.33 |
| ar-fa | 5679.85 | bn-pt | 333.59 | el-ms | 1547.63 | et-fr | 15697.59 | fi-ms | 1276.96 | hi-nl | 92.46 | ja-vi | 679.31 | ml-ru | 310.59 | si-te | 1.85 |
| ar-fi | 17169.90 | bn-ro | 337.94 | el-my | 0.55 | et-fy | 51.63 | fi-nl | 29451.87 | hi-pl | 94.40 | ka-ko | 17.13 | ml-si | 28.01 | si-th | 109.38 |
| ar-fr | 50632.52 | bn-ru | 392.15 | el-ne | 1.04 | et-he | 9814.49 | fi-pl | 29269.50 | hi-pt | 62.44 | ka-lv | 10.71 | ml-sw | 12.47 | si-tl | 3.02 |
| ar-he | 20577.16 | bn-si | 47.49 | el-nl | 37188.78 | et-hi | 43.98 | fi-pt | 681.08 | hi-ru | 142.53 | ka-ms | 31.86 | ml-ta | 15.90 | si-tr | 492.12 |
| ar-hi | 96.26 | bn-sw | 23.91 | el-pl | 35037.31 | et-hu | 16819.43 | fi-ro | 27988.13 | hi-sw | 12.52 | ka-pt | 165.00 | ml-th | 81.03 | si-ur | 4.95 |
| ar-hu | 23770.38 | bn-ta | 15.67 | el-pt | 35491.54 | et-id | 4282.23 | fi-ru | 12403.26 | hi-te | 23.18 | ka-ru | 104.82 | ml-tl | 3.30 | si-vi | 210.15 |
| ar-id | 6989.56 | bn-th | 129.60 | el-ro | 37986.26 | et-it | 14462.11 | fi-si | 391.99 | hi-tl | 0.51 | ka-th | 43.37 | ml-tr | 439.25 | si-zh | 14.28 |
| ar-it | 20070.27 | bn-tl | 2.05 | el-ru | 35037.31 | et-ja | 1176.51 | fi-sw | 29.32 | hi-ur | 176.39 | ka-tr | 178.79 | ml-vi | 124.30 | sw-ta | 6.24 |
| ar-ja | 1847.98 | bn-tr | 441.74 | el-si | 466.44 | et-ka | 110.02 | fi-ta | 20.08 | hi-vi | 101.10 | ka-ur | 1.98 | ml-zh | 34.77 | sw-th | 6.24 |
| ar-ka | 161.65 | bn-ur | 108.74 | el-sw | 171.65 | et-kk | 1.14 | fi-te | 17.13 | hi-zh | 25.57 | ka-zh | 6.52 | ms-nl | 1409.07 | sw-tr | 91.95 |
| ar-kk | 1.28 | bn-vi | 219.57 | el-ta | 125.96 | et-ko | 492.79 | fi-th | 2288.65 | hu-id | 7253.81 | kk-lv | 1.13 | ms-pt | 1523.57 | sw-ur | 50.29 |
| ar-ko | 1262.60 | bn-zh | 85.24 | el-te | 18.10 | et-lt | 7431.17 | fi-tr | 22551.99 | hu-it | 33513.06 | kk-ms | 1.12 | ms-ro | 1732.68 | sw-yo | 0.03 |
| ar-lt | 1177.67 | cs-de | 24049.84 | el-th | 2858.53 | et-lv | 6728.85 | fi-ur | 19.43 | hu-ja | 1767.63 | kk-nl | 1.85 | ms-ru | 1210.56 | sw-zh | 19.31 |
| ar-lv | 433.66 | cs-el | 35372.28 | el-tl | 7.44 | et-sw | 4.85 | fi-vi | 2517.08 | hu-ka | 165.84 | kk-pl | 77.88 | ms-si | 204.06 | ta-te | 21.16 |
| ar-ml | 348.33 | cs-en | 54470.47 | el-tr | 32797.28 | et-ta | 20.44 | fi-zh | 630.12 | hu-ko | 1168.66 | kk-ru | 2.22 | ms-ta | 15.24 | ta-th | 14.15 |
| ar-ms | 1555.33 | cs-es | 44962.42 | el-ur | 7.44 | et-te | 18.10 | fr-he | 21218.88 | hu-lv | 6776.32 | kk-tr | 2.59 | ms-th | 413.17 | ta-tr | 77.76 |
| ar-my | 0.18 | cs-et | 17819.46 | el-vi | 201.28 | et-th | 2505.71 | fr-hi | 46.14 | hu-ms | 1581.43 | kk-vi | 1.18 | ms-tr | 1754.22 | ta-ur | 49.89 |
| ar-ne | 0.41 | cs-eu | 686.53 | el-zh | 649.81 | et-tl | 10.13 | fr-hu | 37027.57 | hu-my | 0.06 | ko-lt | 148.54 | ms-vi | 851.69 | ta-vi | 12.65 |
| ar-nl | 21273.74 | cs-fa | 5417.48 | en-es | 156560.00 | et-tr | 11408.82 | fr-id | 6630.25 | hu-nl | 33904.34 | ko-lv | 57.10 | ms-zh | 85.86 | ta-zh | 13.02 |
| ar-pl | 24819.83 | cs-fi | 28031.47 | en-et | 22284.30 | et-ur | 19.52 | fr-it | 41162.37 | hu-pl | 39869.14 | ko-ml | 42.92 | my-nl | 0.10 | te-th | 0.96 |
| ar-pt | 20379.56 | cs-fr | 34876.02 | en-eu | 805.78 | et-vi | 2048.37 | fr-ja | 1608.52 | hu-pt | 31715.19 | ko-ms | 291.25 | my-pt | 0.10 | te-tr | 18.84 |
| ar-ro | 26187.15 | cs-he | 24503.29 | en-fi | 42783.36 | et-zh | 405.30 | fr-ka | 139.63 | hu-ro | 38807.61 | ko-ne | 16.07 | my-ro | 0.03 | te-vi | 9.34 |
| ar-ru | 45992.72 | cs-hi | 86.86 | en-fr | 161519.91 | eu-fa | 245.78 | fr-kk | 1.34 | hu-ru | 19172.99 | ko-nl | 1120.75 | my-ru | 0.81 | th-tl | 7.28 |
| ar-si | 483.96 | cs-hu | 39272.92 | en-fy | 126.19 | eu-fi | 581.61 | fr-ko | 991.60 | hu-si | 460.99 | ko-pt | 1119.49 | my-sw | 0.15 | th-tr | 3054.07 |
| ar-sw | 16.52 | cs-id | 7310.27 | en-gd | 47.02 | eu-he | 566.71 | fr-lv | 8569.67 | hu-ta | 20.63 | ko-ru | 959.46 | my-tr | 0.03 | th-ur | 58.65 |
| ar-ta | 37.15 | cs-it | 33935.96 | en-he | 30028.28 | eu-hi | 9.98 | fr-ml | 278.47 | hu-th | 2867.23 | ko-si | 58.66 | my-ur | 0.02 | th-vi | 672.82 |
| ar-te | 19.33 | cs-ja | 1806.97 | en-hi | 9677.33 | eu-hu | 663.68 | fr-ms | 1423.08 | hu-tl | 10.79 | | | my-zh | 0.13 | th-zh | 133.45 |
| ar-th | 2959.96 | cs-ka | 163.35 | en-hu | 55233.87 | eu-id | 307.85 | fr-my | 1.47 | hu-tr | 32494.90 | | | ne-nl | 0.09 | tl-tr | 14.51 |
| ar-tl | 7.58 | cs-kk | 1.26 | en-id | 76257.21 | eu-it | 568.66 | fr-ne | 1.45 | hu-ur | 23.32 | | | ne-pt | 0.38 | tl-vi | 5.86 |
| ar-tr | 26683.62 | cs-ko | 1199.62 | en-it | 96257.21 | eu-ja | 139.14 | fr-nl | 47363.70 | hu-vi | 2974.61 | | | ne-ro | 0.04 | tr-ur | 473.08 |
| ar-ur | 126.33 | cs-lt | 7694.12 | en-ja | 2177.89 | eu-ka | 9.42 | fr-pt | 42850.13 | id-it | 5831.16 | | | ne-ru | 1.30 | tr-vi | 3178.03 |
| ar-vi | 2875.00 | cs-lv | 6745.84 | en-ka | 199.98 | eu-ko | 72.17 | fr-ro | 37249.80 | id-ja | 1271.31 | | | ne-sw | 0.05 | tr-zh | 1029.21 |
| ar-yo | 0.01 | cs-ml | 319.93 | en-kk | 3.71 | eu-lt | 108.12 | fr-ru | 54231.81 | id-ka | 85.07 | | | ne-tr | 0.03 | ur-vi | 12.52 |
| ar-zh | 28120.22 | cs-ms | 1592.17 | en-ko | 1493.95 | eu-lv | 36.81 | fr-si | 393.48 | id-ko | 605.78 | | | ne-ur | 0.06 | ur-zh | 99.78 |
| bg-bn | 310.12 | cs-my | 0.08 | en-lt | 10992.89 | eu-ml | 42.72 | fr-sw | 17.57 | id-lv | 342.36 | | | ne-zh | 0.01 | vi-zh | 148.22 |
| bg-cs | 34502.46 | cs-ne | 0.07 | en-lv | 9883.08 | eu-ms | 129.20 | fr-ta | 24.03 | id-ml | 230.67 | | | nl-ro | 36051.60 | | |
| bg-de | 19852.81 | cs-nl | 34427.07 | en-ml | 573.95 | eu-nl | 619.88 | fr-tl | 10.79 | id-ms | 1614.63 | | | nl-ru | 16582.78 | | |
| bg-el | 32130.86 | cs-pt | 32469.01 | en-ms | 2050.83 | eu-pt | 641.30 | fr-th | 2325.22 | id-my | 0.11 | | | nl-si | 410.92 | | |
| bg-en | 47247.04 | cs-ro | 39226.31 | en-my | 2.43 | eu-ro | 715.99 | fr-tr | 29245.91 | id-ne | 0.07 | | | nl-sw | 31.38 | | |
| bg-es | 39728.55 | cs-ru | 19703.43 | en-ne | 2.89 | eu-ru | 435.12 | fr-vi | 2752.32 | id-nl | 6493.33 | | | nl-ta | 39.21 | | |
| bg-et | 15188.54 | cs-si | 454.26 | en-nl | 65918.54 | eu-si | 34.56 | fr-yo | 0.12 | | | | | nl-te | 16.00 | | |
| bg-eu | 605.10 | cs-sw | 17.34 | en-pl | 59729.77 | eu-ta | 2.60 | fy-he | 44.06 | | | | | nl-th | 2548.14 | | |
| bg-fa | 4927.53 | cs-ta | 32.81 | en-pt | 61861.36 | eu-te | 0.73 | fy-it | 37.61 | | | | | nl-tl | 8.18 | | |
| bg-fi | 25191.01 | cs-te | 18.72 | en-ro | 60415.46 | eu-th | 80.75 | fy-ja | 37.61 | | | | | nl-tr | 28822.22 | | |
| bg-fr | 30185.98 | cs-th | 2858.53 | en-ru | 65105.13 | eu-tl | 2.60 | fy-pl | 45.83 | | | | | nl-ur | 171.71 | | |
| bg-he | 22887.40 | cs-tl | 7.44 | en-si | 601.16 | eu-tr | 722.77 | fy-pt | 95.81 | | | | | nl-vi | 2748.22 | | |
| bg-hi | 71.38 | cs-tr | 32797.28 | en-sw | 171.65 | eu-ur | 2.01 | fy-ro | 73.99 | | | | | nl-zh | 866.75 | | |
| bg-hu | 34293.44 | cs-ur | 122.87 | en-ta | 125.96 | eu-vi | 201.28 | fy-sw | 0.37 | | | | | | | | |
| bg-id | 7047.21 | cs-vi | 3040.14 | en-te | 27.22 | | | fy-tr | 45.40 | | | | | | | | |
| bg-it | 27649.85 | cs-zh | 894.87 | en-th | 3375.07 | | | | | | | | | | | | |
| bg-ja | 1658.40 | de-el | 30170.64 | en-tl | 16.03 | | | | | | | | | | | | |
| bg-ka | 193.27 | de-en | 83872.47 | | | | | | | | | | | | | | |
| bg-kk | 3.40 | de-es | 41634.80 | | | | | | | | | | | | | | |
| bg-ko | 1056.96 | de-et | 15186.40 | | | | | | | | | | | | | | |
| bg-lt | 5604.11 | de-eu | 534.93 | | | | | | | | | | | | | | |
| bg-lv | 4748.15 | de-fa | 3948.14 | | | | | | | | | | | | | | |
| bg-ml | 283.77 | de-fi | 25753.06 | | | | | | | | | | | | | | |
| bg-ms | 1506.56 | de-fr | 44392.06 | | | | | | | | | | | | | **Total** | 6,421,152.04 |

Table 6: The statistics of bilingual (parallel) pre-training corpus.

| Pre-training Hyperparameters | Large | Small |
|---|---|---|
| Number of layers | 24 | 6 |
| Hidden Size | 1024 | 768 |
| FFN inner hidden size | 4096 | 3072 |
| Attention heads | 16 | 12 |
| Attention head size | 64 | 64 |
| Embedding Size | 1024 | 768 |
| Mask percent (monolingual/ bilingual) | 15%/25% | 15%/25% |
| Learning Rate Decay | Linear | Linear |
| Warmup steps | 12k | 12k |
| Learning Rate | 2e-4 | 3e-4 |
| Adam $\epsilon$ | 1e-6 | 1e-6 |
| Adam $\beta_1$ | 0.9 | 0.9 |
| Adam $\beta_2$ | 0.98 | 0.999 |
| Attention Dropout | 0.1 | 0.1 |
| Dropout | 0.1 | 0.1 |
| Weight Decay | 0.01 | 0.01 |
| Max Sequence Length (monolingual/bilingual) | 512/128 | 512/128 |
| Batch Size (monolingual/bilingual) | 1024/4096 | 1024/4096 |
| Train Steps | 240k | 240k |
| Total Parameters | 662M | 247M |

Table 7: The pre-training hyperparameters.

| Model | Architecture | Params | Enc Layers | Dec Layers | #Languages | Vocab Size |
|---|---|---|---|---|---|---|
| mBERT (Devlin et al., 2019) | Encoder-only | 110M | 12 | - | 104 | 110k |
| XLM (Lample & Conneau, 2019) | Encoder-only | 570M | 24 | - | 100 | 200k |
| XLM-R (Conneau et al., 2019) | Encoder-only | 550M | 24 | - | 100 | 250k |
| MMTE (Siddhant et al., 2020) | Encoder-decoder | 375M | 6 | 6 | 103 | 64k |
| mBART (Liu et al., 2020b) | Encoder-decoder | 680M | 12 | 12 | 25 | 250k |
| VECO | Variable | 662M | 24* | | 50 | 250k |

Table 8: Comparison of large cross-lingual models. * denotes encoder and decoder share all self-attention and feed-forward sub-layers.

The full set of fine-tuning hyperparameters is listed in Table 9.

| | |
|---|---|
| Learning Rate | [1e-5,2e-5,3e-5] for *Cross-lingual Transfer* |
| | [3e-6,5e-6,1e-5] for *Translate-Train-All* |
| Adam $\epsilon$ | 1e-8 |
| Adam $\beta_1$ | 0.9 |
| Adam $\beta_2$ | 0.999 |
| Batch Size | [16,32,64] |
| Train Epochs | [3,5,10] |

Table 9: The fine-tuning hyperparameters

## C DETAILED RESULTS ON XTREME

The detailed results of each XTREME task under the cross-lingual transfer and translate-train-all settings on all languages are listed in the following tables.

| Model | en | ar | bg | de | el | es | fr | hi | ru | sw | th | tr | ur | vi | zh | **Avg.** |
|---|---|---|---|---|---|---|---|---|---|---|---|---|---|---|---|---|
| *Cross-lingual Transfer* | | | | | | | | | | | | | | | | |
| mBERT[†] | 80.8 | 64.3 | 68.0 | 70.0 | 65.3 | 73.5 | 73.4 | 58.9 | 67.8 | 49.7 | 54.1 | 60.9 | 57.2 | 69.3 | 67.8 | 65.4 |
| MMTE[†] | 79.6 | 64.9 | 70.4 | 68.2 | 67.3 | 71.6 | 69.5 | 63.5 | 66.2 | 61.9 | 66.2 | 63.6 | 60.0 | 69.7 | 69.2 | 67.5 |
| XLM[†] | 82.8 | 66.0 | 71.9 | 72.7 | 70.4 | 75.5 | 74.3 | 62.5 | 69.9 | 58.1 | 65.5 | 66.4 | 59.8 | 70.7 | 70.2 | 69.1 |
| XLM-R[†] | **88.7** | 77.2 | 83.0 | 82.5 | 80.8 | 83.7 | 82.2 | 75.6 | 79.1 | 71.2 | **77.4** | 78.0 | **71.7** | 79.3 | 78.2 | 79.2 |
| VECO | 88.2 | **79.2** | **83.1** | **82.9** | **81.2** | **84.2** | **82.8** | **76.2** | **80.3** | 74.3 | 77.0 | **78.4** | 71.3 | **80.4** | **79.1** | **79.9** |
| *Translate-Train-All* | | | | | | | | | | | | | | | | |
| XLM[‡] | 85.0 | 80.8 | 81.3 | 80.3 | 79.1 | 80.9 | 78.3 | 75.6 | 77.6 | **78.5** | 76.0 | 79.5 | 72.9 | 72.8 | 68.5 | 77.8 |
| XLM-R[†] | 88.6 | 82.2 | 85.2 | 84.5 | 84.5 | 85.7 | 84.2 | **80.8** | 81.8 | 77.0 | 80.2 | 82.1 | **77.7** | 82.6 | 82.7 | 82.6 |
| VECO | **88.9** | **82.4** | **86.0** | **84.7** | **85.3** | **86.2** | **85.8** | 80.1 | **83.0** | 77.2 | **80.9** | **82.8** | 75.3 | **83.1** | **83.0** | **83.0** |

Table 10: XNLI accuracy scores for each language. "[†]" and [‡] results are provided by Hu et al. (2020) and Conneau et al. (2019), respectively.

| Model | en | de | es | fr | ja | ko | zh | **Avg.** |
|---|---|---|---|---|---|---|---|---|
| *Cross-lingual Transfer* | | | | | | | | |
| mBERT | 94.0 | 85.7 | 87.4 | 87.0 | 73.0 | 69.6 | 77.0 | 81.9 |
| XLM | 94.0 | 85.9 | 88.3 | 87.4 | 69.3 | 64.8 | 76.5 | 80.9 |
| MMTE | 93.1 | 85.1 | 87.2 | 86.9 | 72.0 | 69.2 | 75.9 | 81.3 |
| XLM-R | 94.7 | 89.7 | 90.1 | 90.4 | 78.7 | 79.0 | 82.3 | 86.4 |
| VECO | **96.2** | **91.3** | **91.4** | **92.0** | **81.8** | **82.9** | **85.1** | **88.7** |
| *Translate-Train-All* | | | | | | | | |
| VECO | 96.4 | 93.0 | 93.0 | 93.5 | 87.2 | 86.8 | 87.9 | 91.1 |

Table 11: PAWS-X accuracy scores for each language.

| Model | de | fr | ru | zh | **Avg.** |
|---|---|---|---|---|---|
| *Cross-lingual Transfer* | | | | | |
| mBERT | 62.5 | 62.6 | 51.8 | 50.0 | 56.7 |
| XLM | 56.3 | 63.9 | 60.6 | 46.6 | 56.8 |
| MMTE | 67.9 | 63.9 | 54.3 | 53.3 | 59.8 |
| XLM-R | 67.5 | 66.5 | 73.5 | 56.7 | 66.0 |
| VECO | **89.6** | **84.6** | **87.4** | **78.5** | **85.0** |
| *Translate-Train-All* | | | | | |
| VECO | 93.0 | 88.7 | 89.9 | 85.7 | 89.3 |

Table 12: BUCC results (F1 scores).

| Model | en | ar | de | el | es | hi | ru | th | tr | vi | zh | **Avg.** |
|---|---|---|---|---|---|---|---|---|---|---|---|---|
| *Cross-lingual Transfer* | | | | | | | | | | | | |
| mBERT | 83.5 / 72.2 | 61.5 / 45.1 | 70.6 / 54.0 | 62.6 / 44.9 | 75.5 / 56.9 | 59.2 / 46.0 | 71.3 / 53.3 | 42.7 / 33.5 | 55.4 / 40.1 | 69.5 / 49.6 | 58.0 / 48.3 | 64.5 / 49.4 |
| XLM | 74.2 / 62.1 | 61.4 / 44.7 | 66.0 / 49.7 | 57.5 / 39.1 | 68.2 / 49.8 | 56.6 / 40.3 | 65.3 / 48.2 | 35.4 / 24.5 | 57.9 / 41.2 | 65.8 / 47.6 | 49.7 / 39.7 | 59.8 / 44.3 |
| MMTE | 80.1 / 68.1 | 63.2 / 46.2 | 68.8 / 50.3 | 61.3 / 35.9 | 72.4 / 52.5 | 61.3 / 47.2 | 68.4 / 45.2 | 48.4 / 35.9 | 58.1 / 40.9 | 70.9 / 50.1 | 55.8 / 36.4 | 64.4 / 46.2 |
| XLM-R | 86.5 / 75.7 | 68.6 / 49.0 | **80.4 / 63.4** | **79.8 / 61.7** | **82.0 / 63.9** | 76.7 / 59.7 | **80.1 / 64.3** | 74.2 / 62.8 | **75.9 / 59.3** | 79.1 / 59.0 | 59.3 / 50.0 | 76.6 / 60.8 |
| VECO | **87.6 / 76.5** | **73.6 / 56.1** | 79.8 / 62.2 | 79.6 / 61.6 | 81.2 / 61.6 | 74.7 / 57.6 | 78.7 / 62.1 | 72.8 / 60.6 | 75.1 / 58.3 | **79.0 / 59.8** | **69.2 / 59.2** | **77.3 / 61.8** |
| *Translate-Train-All* | | | | | | | | | | | | |
| VECO | 88.3/77.9 | 76.9/61.1 | 80.5/64.6 | 81.5/64.1 | 84.2/66.8 | 78.8/62.5 | 80.2/66.1 | 77.0/70.4 | 77.8/62.2 | 82.5/63.7 | 71.6/69.4 | 79.9/66.3 |

Table 13: XQuAD results (F1 / EM) for each language.

| Model | en | ar | de | es | hi | vi | zh | **Avg.** |
|---|---|---|---|---|---|---|---|---|
| *Cross-lingual Transfer* | | | | | | | | |
| mBERT | 80.2 / 67.0 | 52.3 / 34.6 | 59.0 / 43.8 | 67.4 / 49.2 | 50.2 / 35.3 | 61.2 / 40.7 | 59.6 / 38.6 | 61.4 / 44.2 |
| XLM | 68.6 / 55.2 | 42.5 / 25.2 | 50.8 / 37.2 | 54.7 / 37.9 | 34.4 / 21.1 | 48.3 / 30.2 | 40.5 / 21.9 | 48.5 / 32.6 |
| MMTE | 78.5 / – | 56.1 / – | 58.4 / – | 64.9 / – | 46.2 / – | 59.4 / – | 58.3 / – | 60.3 / 41.4 |
| XLM-R | 83.5 / 70.6 | **66.6 / 47.1** | **70.1 / 54.9** | **74.1 / 56.6** | 70.6 / 53.1 | **74.0 / 52.9** | 62.1 / 37.0 | 71.6 / 53.2 |
| VECO | **83.6 / 70.5** | 65.0 / 44.6 | 69.8 / 54.6 | 73.8 / 55.6 | 69.1 / 51.4 | 73.1 / 51.8 | **67.3 / 43.6** | **71.7 / 53.2** |
| *Translate-Train-All* | | | | | | | | |
| VECO | 84.1/71.3 | 67.8/47.1 | 70.7/55.8 | 74.6/56.6 | 71.1/53.4 | 74.8/54.4 | 68.8/45.8 | 73.1/54.9 |

Table 14: MLQA results (F1 / EM) for each language.

| Model | en | ar | bn | fi | id | ko | ru | sw | te | **Avg.** |
|---|---|---|---|---|---|---|---|---|---|---|
| *Cross-lingual Transfer* | | | | | | | | | | |
| mBERT | **75.3 / 63.6** | 62.2 / 42.8 | 49.3 / 32.7 | 59.7 / 45.3 | 64.8 / 45.8 | **58.8 / 50.0** | 60.0 / 38.8 | 57.5 / 37.9 | 49.6 / 38.4 | 59.7 / 43.9 |
| XLM | 66.9 / 53.9 | 59.4 / 41.2 | 27.2 / 15.0 | 58.2 / 41.4 | 62.5 / 45.8 | 14.2 / 5.1 | 49.2 / 30.7 | 39.4 / 21.6 | 15.5 / 6.9 | 43.6 / 29.1 |
| MMTE | 62.9 / 49.8 | 63.1 / 39.2 | 55.8 / 41.9 | 53.9 / 42.1 | 60.9 / 47.6 | 49.9 / 42.6 | 58.9 / 37.9 | 63.1 / 47.2 | 54.2 / 45.8 | 58.1 / 43.8 |
| XLM-R | 71.5 / 56.8 | 67.6 / 40.4 | **64.0 / 47.8** | 70.5 / 53.2 | **77.4 / 61.9** | 31.9 / 10.9 | **67.0 / 42.1** | **66.1 / 48.1** | 70.1 / 43.6 | 65.1 / 45.0 |
| VECO | 71.3 / 58.2 | **73.1 / 52.8** | 58.9 / 42.5 | **70.9 / 55.1** | 77.2 / 60.0 | 54.2 / 39.9 | 66.1 / 37.6 | 65.8 / 45.7 | **70.6 / 50.7** | **67.6 / 49.1** |
| *Translate-Train-All* | | | | | | | | | | |
| VECO | 77.2/64.8 | 77.0/57.5 | 72.2/56.6 | 76.6/59.3 | 80.0/64.4 | 63.4/52.2 | 72.8/50.5 | 79.4/67.1 | 76.0/58.0 | 75.0/58.9 |

Table 15: TyDiQA-GolP results (F1 / EM) for each language.

| Model | af | ar | bg | de | el | en | es | et | eu | fa | fi | fr | he | hi | hu | id | it |
|---|---|---|---|---|---|---|---|---|---|---|---|---|---|---|---|---|---|
| *Cross-lingual Transfer* | | | | | | | | | | | | | | | | | |
| mBERT | 86.6 | 56.2 | 85.0 | 85.2 | 81.1 | 95.5 | 86.9 | 79.1 | 60.7 | 66.7 | 78.9 | 84.2 | 56.2 | 67.2 | 78.3 | 71.0 | 88.4 |
| XLM | 88.5 | 63.1 | 85.0 | 85.8 | 84.3 | 95.4 | 85.8 | 78.3 | 62.8 | 64.7 | 78.4 | 82.8 | 65.9 | 66.2 | 77.3 | 70.2 | 87.4 |
| MMTE | 86.2 | 65.9 | 87.2 | 85.8 | 77.7 | **96.6** | 85.8 | 81.6 | 61.9 | 67.3 | 81.1 | 84.3 | 57.3 | 76.4 | 78.1 | **73.5** | 89.2 |
| XLM-R | **89.8** | **67.5** | **88.1** | 88.5 | 86.3 | 96.1 | 88.3 | 86.5 | 72.5 | 70.6 | 85.8 | 87.2 | **68.3** | **76.4** | 82.6 | 72.4 | 89.4 |
| VECO | 88.3 | 67.4 | 87.4 | **88.5** | **86.7** | 95.9 | **89.0** | **87.8** | **75.1** | 70.9 | 86.2 | 88.9 | 67.5 | 76.2 | **82.9** | 72.9 | **89.9** |

| | ja | kk | ko | mr | nl | pt | ru | ta | te | th | tl | tr | ur | vi | yo | zh | Avg. |
|---|---|---|---|---|---|---|---|---|---|---|---|---|---|---|---|---|---|
| mBERT | **49.2** | 70.5 | 49.6 | 69.4 | 88.6 | 86.2 | 85.5 | 59.0 | 75.9 | 41.7 | 81.4 | 68.5 | 57.0 | 53.2 | **55.7** | 61.6 | 71.5 |
| XLM | 49.0 | 70.2 | 50.1 | 68.7 | 88.1 | 84.9 | 86.5 | 59.8 | 76.8 | **55.2** | 76.3 | 66.4 | 61.2 | 52.4 | 20.5 | 65.4 | 71.3 |
| MMTE | 48.6 | 70.5 | **59.3** | 74.4 | 83.2 | 86.1 | 88.1 | 63.7 | 81.9 | 43.1 | 80.3 | 71.8 | 61.1 | 56.2 | 51.9 | **68.1** | 73.5 |
| XLM-R | 15.9 | 78.1 | 53.9 | 80.8 | 89.5 | 87.6 | 89.5 | **65.2** | **86.6** | 47.2 | 92.2 | 76.3 | **70.3** | 56.8 | 24.6 | 25.7 | 73.8 |
| VECO | 31.4 | **79.3** | 53.1 | **84.3** | **89.8** | **88.3** | **90.2** | 64.3 | 85.8 | 48.0 | **93.7** | **77.2** | 69.2 | **58.1** | 26.2 | 39.4 | **75.1** |

Table 16: POS results (Accuracy) for each language.

| Model | en | af | ar | bg | bn | de | el | es | et | eu | fa | fi | fr | he | hi | hu | id | it | ja | jv |
|---|---|---|---|---|---|---|---|---|---|---|---|---|---|---|---|---|---|---|---|---|
| *Cross-lingual Transfer* | | | | | | | | | | | | | | | | | | | | |
| mBERT | **85.2** | 77.4 | 41.1 | 77.0 | 70.0 | 78.0 | 72.5 | 77.4 | 75.4 | 66.3 | 46.2 | 77.2 | 79.6 | 56.6 | 65.0 | 76.4 | **53.5** | 81.5 | 29.0 | **66.4** |
| XLM | 82.6 | 74.9 | 44.8 | 76.7 | 70.0 | 78.1 | 73.5 | 74.8 | 74.8 | 62.3 | 49.2 | 79.6 | 78.5 | **57.7** | 66.1 | 76.5 | 53.1 | 80.7 | 23.6 | 63.0 |
| MMTE | 77.9 | 74.9 | 41.8 | 75.1 | 64.9 | 71.9 | 68.3 | 71.8 | 74.9 | 62.6 | 45.6 | 75.2 | 73.9 | 54.2 | 66.2 | 73.8 | 47.9 | 74.1 | **31.2** | 63.9 |
| XLM-R | 84.7 | **78.9** | **53.0** | 81.4 | **78.8** | 78.8 | **79.5** | **79.6** | 79.1 | 60.9 | 61.9 | 79.2 | **80.5** | 56.8 | **73.0** | 79.8 | 53.0 | 81.3 | 23.2 | 62.5 |
| VECO | 83.8 | 77.5 | 48.2 | **83.9** | 77.2 | **79.4** | 79.3 | 75.4 | **80.4** | **68.3** | **68.2** | **80.6** | 80.1 | 55.0 | 71.0 | **80.9** | 52.9 | **81.7** | 19.4 | 63.2 |

| | ka | kk | ko | ml | mr | ms | my | nl | pt | ru | sw | ta | te | th | tl | tr | ur | vi | yo | zh |
|---|---|---|---|---|---|---|---|---|---|---|---|---|---|---|---|---|---|---|---|---|
| mBERT | 64.6 | 45.8 | 59.6 | 52.3 | 58.2 | **72.7** | 45.2 | 81.8 | 80.8 | 64.0 | 67.5 | 50.7 | 48.5 | 3.6 | 71.7 | 71.8 | 36.9 | 71.8 | **44.9** | **42.7** |
| XLM | 67.7 | **57.2** | 26.3 | 59.4 | 62.4 | 69.6 | 47.6 | 81.2 | 77.9 | 63.5 | 68.4 | 53.6 | 49.6 | 0.3 | **78.6** | 71.0 | 43.0 | 70.1 | 26.5 | 32.4 |
| MMTE | 60.9 | 43.9 | 58.2 | 44.8 | 58.5 | 68.3 | 42.9 | 74.8 | 72.9 | 58.2 | 66.3 | 48.1 | 46.9 | **3.9** | 64.1 | 61.9 | 37.2 | 68.1 | 32.1 | 28.9 |
| XLMR | **71.6** | 56.2 | **60.0** | 67.8 | 68.1 | 57.1 | 54.3 | **84.0** | 81.9 | 69.1 | 70.5 | 59.5 | 55.8 | 1.3 | 73.2 | 76.1 | 56.4 | **79.4** | 33.6 | 33.1 |
| VECO | 67.1 | 51.2 | 59.9 | 63.4 | 65.0 | 70.0 | **56.1** | 83.4 | **83.1** | 71.3 | 70.5 | 60.5 | 56.2 | 1.4 | 71.3 | **80.4** | **69.3** | 76.0 | 37.4 | 29.1 |

Table 17: NER results (F1) for each language.

| Model | af | ar | bg | bn | de | el | es | et | eu | fa | fi | fr | he | hi | hu | id | it | ja |
|---|---|---|---|---|---|---|---|---|---|---|---|---|---|---|---|---|---|---|
| *Cross-lingual Transfer* | | | | | | | | | | | | | | | | | | |
| BERT | 42.7 | 25.8 | 49.3 | 17 | 77.2 | 29.8 | 68.7 | 29.3 | 25.5 | 46.1 | 39 | 66.3 | 41.9 | 34.8 | 38.7 | 54.6 | 58.4 | 42 |
| XLM | 43.2 | 18.2 | 40 | 13.5 | 66.2 | 25.6 | 58.4 | 24.8 | 17.1 | 32.2 | 32.2 | 54.5 | 32.1 | 26.5 | 30.1 | 45.9 | 56.5 | 40 |
| XLMR | **58.2** | 47.5 | 71.6 | 43 | 88.8 | 61.8 | 75.7 | 52.2 | 35.8 | 70.5 | 71.6 | 73.7 | 66.4 | 72.2 | 65.4 | 77 | 68.3 | 60.6 |
| VECO | 48.2 | 70.9 | 86.7 | 57.7 | 97.5 | 81.5 | 94.8 | 89.7 | 62.9 | 82.1 | 87.9 | 88.8 | 74.7 | 80.7 | 87.6 | 89.6 | 89.2 | 83.2 |
| *Translate-Train-All* | | | | | | | | | | | | | | | | | | |
| VECO | 80.9 | 85.1 | 91.3 | 78.1 | 98.5 | 89.5 | 97.4 | 94.8 | 79.8 | 93.1 | 95.4 | 93.7 | 85.8 | 94.2 | 93.8 | 93.0 | 92.2 | 92.8 |

| | jv | ka | kk | ko | ml | mr | nl | pt | ru | sw | ta | te | th | tl | tr | ur | vi | zh |
|---|---|---|---|---|---|---|---|---|---|---|---|---|---|---|---|---|---|---|
| *Cross-lingual Transfer* | | | | | | | | | | | | | | | | | | |
| BERT | 17.6 | 20.5 | 27.1 | 38.5 | 19.8 | 20.9 | 68 | 69.9 | 61.2 | 11.5 | 14.3 | 16.2 | 13.7 | 16 | 34.8 | 31.6 | 62 | 71.6 |
| XLM | **22.4** | 22.9 | 17.9 | 25.5 | 20.1 | 13.9 | 59.6 | 63.9 | 44.8 | 12.6 | 20.2 | 12.4 | 31.8 | 14.8 | 26.2 | 18.1 | 47.1 | 42.2 |
| XLMR | 14.1 | 52.1 | 48.5 | 61.4 | 65.4 | 56.8 | 80.8 | 82.2 | 74.1 | 20.3 | 26.4 | 35.9 | 29.4 | 36.7 | 65.7 | 24.3 | 74.7 | 68.3 |
| VECO | 17.6 | **58.5** | 53.9 | 75.3 | 80.1 | 64.2 | 94.4 | 92.8 | 88.6 | 37.4 | 61.9 | 65.8 | 84.5 | 52.5 | 89.3 | 64.3 | 85.8 | 82.7 |
| *Translate-Train-All* | | | | | | | | | | | | | | | | | | |
| VECO | 35.1 | 83.0 | 74.1 | 88.7 | 94.8 | 82.5 | 95.9 | 94.6 | 92.2 | 69.7 | 82.4 | 91.0 | 94.7 | 73.0 | 95.2 | 63.8 | 95.1 | 93.9 |

Table 18: Tatoeba results (Accuracy) for each language

