# OpenReview forum: "VECO: Variable Encoder-decoder Pre-training for Cross-lingual Understanding and Generation"
_ICLR.cc/2021/Conference — Reject_

### Official Review · AnonReviewer2 · 2020-10-28
**A multilingual pre-trained model with shared Self-Attention and FFN weights**

**Rating:** 5
**Confidence:** 4

**Review:**

This paper targets to unify the advantages of encoder-only model and encoder-decoder model for multilingual pre-training. Given three types of parameters in transformer blocks including Self-Attention, Cross-Attention and FFN, this paper proposes to train the model for understanding and generation tasks at the same time, where Self-Attention and FFN weights are shared and trained for both understanding and generation tasks and Cross-Attention weights are trained for generation tasks only. Evaluations are performed on XTREME and WMT datasets, respectively, with good improvements obtained.

Strengths: (1) The approaches are clear and easy to reproduce. (2) The performance on XTREME and WMT are good.

Weakness: (1) Based on my understanding, this approach equals to sharing Self-Attention and FFN weights between MLM encoder, MT encoder and MT decoder, which is not innovative enough. (2) More baseline settings could be added for a more solid comparison. For example, for XTREME, what's the performance of IS-MLM + TLM, which is pre-trained using the full training corpus? (3) what kind of training data is used in the CS-MLM task? I cannot find the data detail from the paper. Is  it the MT data? (4) Do Table 1 and Table 2 use the same training corpus?

- Additional question. (1) For task IS-MLM and CS-MLM, do they use both monolingual data and bilingual data? If yes, which type of data is more useful?  (2) In table 2, do XLM and VECO use the same vocabulary and the same code for MLM? (3) If the answer to question (2) is yes, then the first line (XLM_{SMALL}) in Table 2 corresponds to XLM-R in Table 1 and the fourth line (VECO_{SMALL}) corresponds to VECO in table 2. For XNLI, the gap between VECO_{SMALL} and XLM_{SMALL} in Table 2 is 7.9, but the gap between XLM_R and VECO in Table 1 is only 0.7. What caused this difference? (4) In Table 3, different models use different hyperparameter settings, especially encoder/decoder with different layers. Could the paper use the same layer setting when compare with a specific baseline? This would help to verify that the gain comes from the new pre-training strategy, instead of a deeper model.

- Suggestions: (1) Add more baselines and ablation study, this will help readers to understand where the gains come from. For example, add a baseline model pre-trained by XLM-R and TLM in Table 1. This could prove the gain of VECO is not from TLM. (2) XGLUE, a concurrent work of XTREMEM, includes both multilingual understanding and generation tasks. I suggest the paper can provide the results of VECO on the multilingual NLG tasks in XGLUE.

---

> ### Author Response · Authors · 2020-11-19
> **Response to AnonReviewer2**
>
> Thank you for the comments! We address some of the concerns and questions below:
>
> **2-1: "This approach equals to sharing Self-Attention and FFN weights between MLM encoder, MT encoder and MT decoder"**
>
> The core contribution of this article is the exquisite cooperation of parameters sharing and pre-training strategy makes it flexibly initialize any framework. Please refer to responses **4-3** and **5-1** for more details.
>
> **2-2: “For XTREME, what's the performance of IS-MLM + TLM, which is pre-trained using the full training corpus?” and “add a baseline model pre-trained by XLM-R and TLM in Table 1”**
>
> There might be some misunderstanding regarding your above two questions. Note that it is really time-consuming and costly to do ablation studies in a large-sized model setting in Table 1. As an alternative, we already provide a small-sized baseline in Table 2 (third row) which is pre-trained using IS-MLM + TLM tasks under XLM/XLM-R framework. Comparing the third row and fourth row of Table 2, we can conclude that the gain of VECO is not from TLM. Besides, if you are wondering why we don’t release  the results of other 8 tasks in XTREME for baseline in Table 2, it’s because the XNLI is the most widely used datasets in the XTREME. Most papers (e.g., XLM-R) show the main results only on XNLI. It’s not fully necessary to experiment on the other 8 tasks in XTREME for baselines, since the gain of VECO on XNLI is already excellent.
>
> **2-3: “What kind of training data is used in the CS-MLM? Is it the MT data? Do Table 1 and Table 2 use the same training corpus?”**
>
> All mentioned above can be found in our submission version. In the first paragraph of Section 2.2, we mentioned that the CS-MLM task uses a pair of input (x, y), which is composed of “two adjacent segments in the monolingual corpus and a pair of parallel sentence in the bilingual (parallel) corpus”. In Section 3 and Appendix A, we have shown the detailed information of monolingual and bilingual corpus. Table 2 (small-sized models) only uses a “subset” of full training data in Table 1, mentioned in the last paragraph of Section 4.2 and Appendix A. Please let us know if there are other details you find unclear.
>
> **2-4: “In table 2, do XLM and VECO use the same vocabulary and the same code for MLM?”, “the gap between VECO_{SMALL} and XLM_{SMALL} in Table 2 is 7.9, but the gap between XLM_R and VECO in Table 1 is only 0.7 What caused this difference?”**
>
> All the models in Table 2 are trained from scratch by ourselves, using the same vocabulary, code, and data. So the comparison in Table 2 is totally comparable. However, the results in Table 1 are not totally comparable, since the models are released by the original papers, and many factors that can affect training results. That's why we did a comparable experiment on small-sized models in Table 2. Meanwhile, even though all the settings are the same, effective pre-training tasks can achieve much more absolute gains on the small models than large models. For example, ELECTRA shows that the gap between small-sized ELECTRA and BERT is 4.8 on GLUE, while the gap of large-sized models narrows to 1.8. Under a cross-lingual scenario, the gap can be larger because there exists a balance of different languages. More specifically, the small model is more urgent to deal with language balance problem due to the limited size, thus let the effective pre-training tasks (especially CS-MLM which train a cross-attention to enhance the cross-lingual mapping) play a greater role.
>
> **2-5: “Could the paper use the same layer setting in Table 3”**
>
> The reason why we didn’t use the totally same layer setting is caused by the characteristic of pre-trained models: (1) The layer of pre-trained encoder-decoder models decide the layer of MT experiments. Thus the layer of mBART is set the same as pre-training (i.e., 12-layer encoder and 12-layer decoder); (2) The layer of pre-trained encoder models only decides the layer of the MT encoder, while the decoder layer can be any value. Thus XLM-R initialized model is fixed as a 24-layer encoder. In order to minimize the layer gap between XLM-R and mBART initialized MT model, we choose the most common 6-layer decoder in Table 3 and 3-layer decoder in Table 4. To have a fair comparison to XLM-R, VECO also adopts the same layer settings.
> In conclusion, we tried our best to use the same layer setting among most of the models if possible (Transformer-xBig, XLM-R, and VECO), which can basically verify that the gain comes from the new pre-training strategy, instead of a deeper model. And VECO can even outperform a most recent deeper randomly initialized Transformer (60-layer encoder + 12-layer decoder), which can further verify that improvement is from a deeper setting.
>
> **2-6: “Provide the results of VECO on the multilingual NLG tasks in XGLUE.”**
>
> Since there are only two participants on the XGLUE, we choose to experiment on WMT. We are working on adding XGLUE results.

---

### Official Review · AnonReviewer3 · 2020-10-28
**Official Blind Review #3**

**Rating:** 7
**Confidence:** 4

**Review:**

This paper proposes a **unified** cross-lingual pretraining method that works well for both natural language *understanding* (NLU)---typically done using *encoder-only* architectures like mBERT and XLM---and conditional natural language *generation* (NLG) tasks like machine translation---typically done using *encoder-decoder* architectures like mBART. More concretely, let $\mathbf{x}$ be the "source" sequence and $\mathbf{y}$ the "target" sequence. In machine translation, $\mathbf{x}$ and $\mathbf{y}$ are the source and target sentences, while in the monolingual case, $\mathbf{x}$ and $\mathbf{y}$ are two contiguous sequences in the corpus. The pretraining loss consists of the following four terms (Eq. 8): (i) masked language modelling only on $\mathbf{x}$, (ii) masked language modelling only on $\mathbf{y}$, (iii) autoregressively decoding $\mathbf{y}$ conditional on $\mathbf{x}$ in a left-to-right fashion, and (iv) the reverse of the third loss term, i.e. autoregressively decoding $\mathbf{x}$ given $\mathbf{y}$ in a left-to-right fashion.

Compared to prior approaches, this proposed approach has a key benefit of *sharing* the parameters between the encoder and the decoder during pretraining (hence eliminating the need to train a separate decoder component from scratch), and also enables the pretrained model to be used for both NLU (where the unused cross-attention module is simply discarded) and NLG tasks. Experiments demonstrate the strong empirical performance of the proposed approach on both cross-lingual NLU (through the XTREME benchmark) and two machine translation datasets, surpassing the previous state of the art results in many cases. Further analysis suggests that the improvements are not simply due to training on more bilingual data.

**Pros:**
1. The proposed approach demonstrates strong empirical performance on both cross-lingual NLU and machine translation tasks, outperforming various strong baselines and achieving new state of the art results in many cases.

2. The paper does a great job of analysing whether the improvements come from the proposed approach---which benefits from the encoder-decoder parameter sharing and a loss function that takes into account both intra-sequence masked language modelling and sequence-to-sequence mapping---or simply from more bilingual data. Disentangling the gains from the proposed approach vs more data is really important to understand and enable better progress in the field, which unfortunately is not always done in prior work. So I appreciate the fact that this paper takes a step towards addressing this question, and it is encouraging that the improvements do not merely come from more data (Table 2).

3. The paper is overall well-written and draws extensive connections to the relevant prior work.

**Cons:**
1. In my understanding, the difference between the proposed approach and mBART (which also unifies cross-lingual NLU and NLG) is pretty minimal---the second paragraph of page 4 mentions that the key difference here is the parameter sharing between the encoder and decoder. Could the stronger performance of the model compared to mBART be simply due to training on more bilingual data? Repeating the ablation experiments as done in Table 2, but comparing with mBART rather than only XLM, would help make the empirical setup much stronger.

2. The pretraining objective in Eq. (8) includes a "flipped" sequence-to-sequence mapping term that  predicts $\mathbf{x}$ autoregressively conditional on $\mathbf{y}$. In the monolingual case, this runs the risk of ignoring *discourse information* that comes from the sentence ordering. Imagine a very simple example where $\mathbf{x}=$"Clyde is an elephant." and $\mathbf{y}$="He is very gentle.". The pronoun "He" in $\mathbf{y}$ is allowed since there is already an antecedent, Clyde, in the previous sentence $\mathbf{x}$. But this ceases to be true if we flip the ordering of the sentences. Could you please say more about this?

3. In practice, the model parameters seem to be initialised from XLM-R (except for the cross-attention part that XLM-R does not have). This multi-stage training process can potentially benefit from an implicit model combination effect, which the baseline models may not necessarily be able to benefit from. Having equally positive results where the model is trained from scratch can help make the empirical results stronger.

4. Some of the presentational aspects can be improved. More details below.

**Presentation / grammatical mistakes / typos:**
1. In page 3, "... is that $\mathbf{Q} = \mathbf{K} = \mathbf{V} ...$". I am not sure what this means, since it implies that the matrices $\mathbf{Q, K, V}$ are equivalent.

2. In page 3, the bottom of Eq. (2) says that $\text{MultiHead}(\mathbf{Q}=\mathbf{x}, \mathbf{K}=\mathbf{y}, \mathbf{V}=\mathbf{y})$. I am not sure if this is correct in the standard case where $\mathbf{x}$ is the source sentence and $\mathbf{y}$ is the target sentence. Shouldn't the query $\mathbf{Q}$ come from the target sequence $\mathbf{y}$ while $\mathbf{K}$ and $\mathbf{V}$ come from source sequence $\mathbf{x}$?

3. In page 3, "... and FFN modules to *corporate*" -> "cooperate".

4. At the top of page 4, it is mentioned that "...we detach $\mathbf{X}^{(N)}$...to let the two objectives optimized in isolation". What does "detach" mean here, and how does it relate to optimising the two objectives in isolation?

5. In Table 3, "Randomly *Initialize*" and "Cross-Lingual Models *Initialize*" -> "Initialized".

6. In page 7, "*Figure 4* (left) contrasts two ways ..." -> "Table 4".

7. In page 8, "... and then *continue-train* ..." -> "continue to train".

**Bottom line**
Overall, I think the pros outweigh the cons and this paper can be useful for the community, although addressing some of the concerns I raised above can make the paper stronger. I would be willing to reconsider my score based on the response.

**Update after the authors' response**
The authors have provided a comprehensive response that address most of my comments, and clearly clarified the novelty and key differences with prior work such as mBART (which also seems to be a key concern in the other reviews). Given the satisfactory authors' response, I am therefore raising my score to "7". Overall, I believe this paper presents a good contribution to the field.

---

> ### Author Response · Authors · 2020-11-19
> **Response to AnonReviewer3**
>
> Thank you for the comments! We address some of the concerns and questions below:
>
> **3-1: "The key difference between mBART is the parameter sharing between the encoder and decoder", "Could the stronger performance of the model compared to mBART be simply due to training on more bilingual data?"**
>
> In terms of model architecture, the main difference is that we share the encoder-decoder. However, in terms of task design, we differ in several ways. The proposed IS-MLM task forces the model to bidirectionally comprehend the source input (good for NLU), which is a shortage of mBART. Meanwhile, CS-MLM predicts the masked words other than generating the next word like mBART, thus keeping in line with IS-MLM towards a more consistent optimization direction (predicting masked words) on the shared parameters. Thus, the design of pre-training tasks and parameters sharing *"collectively"* make up the contribution of this work. Moreover, the parameter sharing acts as a form of regularization that stabilizes the training and helps with generalization. Thus, the gains compared to mBART are not purely come from the more bilingual data.  We are working on training a smaller mBART from scratch using both the monolingual and bilingual data (It costs about one more week), since it is really time-consuming and costly to train a large one. As a comparison, we need to add a small-sized VECO experiment on the WMT datasets. Once we finished those experiments, we will add them to the paper or respond to you with the results.
>
> **3-2: "The action that flips sequence-to-sequence mapping term runs the risk of ignoring discourse information that comes from the sentence ordering"**
>
> The situation you said may indeed exist. There are two potential advantages that may overlap the risk from this "flipping" operation:
> 1) One is that flipping the sentence order can create *"harder"* example pairs, thus pushing the model toward a stronger ability of language modeling and understanding. For example, ALBERT shows that using a sentence-order prediction loss will improve the NLU performance.
> 2) Meanwhile, the treatment of (x, y) and (y, x) without discrimination will also improve the robustness of the model among languages, considering the different structure of syntactic and the diverse relationship between sentences.
>
> **3-3: "Initializing from XLM-R can potentially benefit from an implicit model combination effect",  "Having equally positive results where the model is trained from scratch can help make the empirical results stronger."**
>
> First of all, the models in Table 2 are all trained from scratch. Thus we can directly conclude from Table 2 that VECO can outperform XLM/XLM-R when trained on the same data from scratch. Secondly, it is a common practice in pre-training works (e.g., UNILM, Unicoder, InfoXLM) to initialize by BERT/XLM-R to reduce the pre-training time.  This operation may benefit the pre-training process due to a good start point. However, it may also potentially hurt the final performance since it can’t jump out the local optimum of the initialized model. Thus, whether it can largely boost the performance of the initial model depends on the effectiveness of pre-training tasks/strategies compared to the original ones. Table 1 shows large gains when continuing to train  XLM-R using the proposed pre-training tasks. So to speak, the initial model would benefit from our VECO training and achieves better results, rather than the opposite. We are working on training a VECO model from scratch (it just takes lots of compute!), which will help make the above-mentioned claims more convincing.
>
> **3-4: Regarding the Presentation / grammatical mistakes / typos.**
>
> $\mathbf{Q}=\mathbf{K}=\mathbf{V}$ means that the query, key and value are the same hidden states from the previous layer. Thank you for pointing out the minor mistake in Eq. (2). The correct equation is $\mathtt{MultiHead}(\mathbf{Q}=\mathbf{y}, \mathbf{K}=\mathbf{x}, \mathbf{V}=\mathbf{x}$).

---

> > ### Comment · AnonReviewer3 · 2020-11-20
> > **Reply to the Authors' Response**
> >
> > Thank you for the detailed response. The response addresses some of the issues that I raised, and clarifies the difference between the proposed approach and prior work like mBART.
> >
> > Having read: (i) the other reviews and (ii) the authors' responses to those reviews, I believe that the method and findings can be useful for the community. Even though the paper is not particularly groundbreaking in terms of technical novelty, the underlying idea of formulating a unified pretraining method for NLU and NLG in a way that: (i) improves performance on both types of tasks, and (ii) can be easily applicable to both monolingual and multi-lingual (e.g. translation) data is arguably a worthwhile contribution. This is of course in addition to the very strong empirical results, as noted by R1.
> >
> > I will keep an eye on the other ongoing reviewer discussions and read the new version more closely, and then adjust my recommendation accordingly. So far, I think that having this paper presented at the conference would be useful for the community.

---

### Official Review · AnonReviewer4 · 2020-10-29
**Extensive experiments and results, but weak contribution**

**Rating:** 4
**Confidence:** 5

**Review:**

In this paper, the authors propose variable encoder-decoder (VECO), a pre-training strategy for both NLU and NLG tasks. In VECO, two masked language model (MLM) are leveraged for pre-training, (a) inner sentence MLM with encoder only, and (b) cross sentence MLM with both encoder and decoder. The parameters of self-attention and feedforward layers are shared in encoder and decoder. Improvements are observed for multiple NLU and NMT tasks.

The paper is mostly well written and nicely presented. The authors show extensive results in various downstream understanding and generation tasks, improves performances in most cases. The proposed method is simple and straightforward; and can be readily reproduced in any existing toolkit.

I have some doubts on the motivation of separate pre-training tasks for encoder and encoder-decoder. The authors claims that for previous encoder-decoder pre-training like MASS and mBART,
"it usually requires more computation and memory to match the performance of the encoder-only models". What exactly is the additional computation and memory required here? And how is the empirical comparison on these approaches against the proposed one in terms of both model performance and computation/memory cost?
Actually in MMTE [1], the authors show that pre-training a NMT (encoder-decoder) model and extract the encoder is effective for various encoder-only downstream tasks.

Some more comparison with previous pre-training approaches should be presented as well, e.g. MASS, BART, MMTE[1], etc.

In general, I think the authors did an excellent job validating their method on various different NLU/NMT datasets. However, I'm skeptical about the novelty and the general contribution/impact of the paper.

Misc:

* Table 3, why comparing with different number of encoder/decoder layers for previous methods (e.g. 24/6 for XLM-R, 12/12 for mBART)?
* Table 3, do you also share parameters (self-attn, ffn) for baseline methods? How much does parameter sharing contribute in terms of performance and training efficiency?

[1] Siddhant, Aditya, et al. "Evaluating the Cross-Lingual Effectiveness of Massively Multilingual Neural Machine Translation." AAAI. 2020.

---

> ### Author Response · Authors · 2020-11-19
> **Response to AnonReviewer4**
>
> Thank you for the comments! We address some of the concerns and questions below:
>
> **4-1: Regarding the computation and memory cost of utilizing pre-trained encoder-decoder models for NLU downstream tasks.**
>
> There are two ways to apply a pre-trained encoder-decoder to NLU tasks. One is to keep the whole encoder-decoder like BART,  while the other is to only extract the encoder like MMTE.
>
> For BART, the same input is fed into the encoder and decoder and the final output from the decoder is used. The BART model can't beat RoBERTa with comparable training resources. However, BART has extra cross-attention modules (roughly 10%~20% more parameters than encoder-only model).
>
> For MMTE,  it only extracts the encoder while discarding the decoder. The computation/memory cost of MMTE is the same as the encoder-only model. However, MMTE (trained on in-house parallel corpus) only compares to a weak baseline mBERT (only trained on monolingual Wikipedia, other than larger Common Crawl dataset). Thus, it can’t directly conclude from MMTE that extracting the encoder from a pre-trained encoder-decoder is better than training an encoder-only model. We had the same doubts as you in the early part of the VECO study, and we have already done two confirmatory experiments:
> - We used XLM-R to initialize an encoder-decoder model, and then pre-trained with DAE task (for monolingual data) and machine translation task (for parallel data) until convergence. The performance of the encoder dropped by 3.2% on XNLI, compared to the XLM-R encoder used for initialization.
> - We extracted the encoder of mBART and fine-tuned it on XNLI, the results also dropped by more than 5%, compared to utilize the whole encoder-decoder model.
>
> Based on these observations, we claim that “it requires more computation and memory to match the performance of the encoder-only models”.  Please let us know if there are other claims you find unclear.
>
> **4-2: Regarding more comparison with MASS, BART, MMTE.**
>
> It is worth noting that we focus on multilingual pre-training, while BART is monolingual (English) models ( i.e. they are not applicable to cross-lingual tasks). It’s more appropriate to compare with the multilingual version of BART (i.e. mBART). We already compared with mBART in Table 3. Regarding MMTE, we have added the XTREME results to Table 2 (13.6% lower than VECO) in our updated version.
>
> **4-3: Regarding the novelty and the contribution/impact of the paper.**
>
> What I must admit is that the first impression of VECO is only the parameter sharing, due to the writing and formulation. We must clarify that the core contribution of this article exists in the exquisite cooperation of parameters sharing and pre-training tasks: IS-MLM and CS-MLM. IS-MLM forces the model to bidirectionally comprehend the source input (good for NLU), which is a shortage of BART and MMTE since they don’t have any direct constraint (loss) in the encoder. CS-MLM predicts the masked words other than generating the next word in MASS/mBART, thus keeping "a more consistent optimization direction" in line with IS-MLM (predicting masked words) on shared parameters. Thus, the CS-MLM task is totally different from the “shifted” language modeling (generation) task adopted in MASS/BART/mBART.  Most importantly, VECO's inherent **"flexibility"** makes it naturally initialize various Transformer frameworks (e.g., encoder-only, encoder-decoder, even decoder-only), which can benefit a wider range of downstream tasks with the most streamlined parameters. This is also the motivation of why we design "separate pre-training tasks for encoder and encoder-decoder" (as you doubted).
>
> Besides, you mention that we do “excellent experiments" but with “weak contribution”. However, we stand by the idea that one of the most important “contributions” to pre-training works is the remarkable SOTA results. For example, RoBERTa and XLM-R. We hope more people can embrace such seeming “novelty limited” pre-training works with a lot more love and attention, since they indeed largely contribute to the various tasks and faster the research process.
>
> **4-4: "Table 3, why comparing with different number of encoder/decoder layers for previous methods?"**
>
> Please refer to response **2-5** of AnonReviewer2 for reasons.
>
> **4-5: "Table 3, do you also share parameters for baseline methods? How much does parameter sharing contribute in terms of performance and training efficiency?"**
>
> There might be some misunderstandings. We only do parameter sharing during pre-training. When fine-tuning on NLG tasks, we don't share parameters for all models including baselines in Table 3. Regarding that the parameter sharing and pre-training tasks are bound together, the performance is extremely well. Meanwhile, the training efficiency keeps the same as before (due to detach or stop_gradient operation that avoiding training twice deeper networks in CS-MLM). If we misunderstand what you mean, please clarify it. We welcome any response.

---

### Official Review · AnonReviewer1 · 2020-11-02
**Good paper demonstrating strong cross-lingual transfer learning capabilities**

**Rating:** 9
**Confidence:** 5

**Review:**

Summary:

The paper presents VECO: a pre-trained encoder-decoder model which is capable of both cross-lingual understanding and generation. They present two losses for inner-sequence and cross-sequence understanding. These components are then used to either build a encoder-only model or an encoder-decoder model. They present results on XTREME for cross-lingual understanding and on MT for generation. The results on both tasks are quite competitive and clearly shows the benefit of using both monolingual and parallel data with IS_MLM and CS-MLM.


Reasons for score:

I vote for accepting the paper. The paper initializes from XLM-R and then fine-tunes it on monolingual and parallel data in 50 languages. The authors get +3 average gain over the previous best system on XTREME and were ranked #1 at the time of submission. They also get handy gains in the MT benchmarks. The two ablation experiments show that the new CS-MLM task is indeed beneficial and improves the performance. One of the concerns I have is that the authors are not upfront about their model being in just 50 languages and hence might not be comparable to other models like XLM-R which support 100+ languages.x


Cons:
- As mentioned above, please be upfront about training only on 50 languages. It's only mentioned in the Appendix. This needs to be mentioned in the main text and maybe even in the results table.
- I would like to see more information about the amount of data (both monolingual and parallel) listed clearly in the Appendix.
- What would happen if VECO was trained on 100 languages from XLM-R? That result would be interesting to see in Table 1.
- I would personally like to see more ablation experiments: experiments where the amount of pre-trained monolingual and parallel data were reduced independently to see the impact it has performance.


Minor comments:
In Section 4.1, kindly cite all the representative tasks in XTREME as suggested here:
https://github.com/google-research/xtreme#paper

- It would be great if the authors stated the number of monolingual and parallel examples explicitly in Section 3. Appendix A doesn't provide the number of examples but just the size (1TB) of parallel data used.

- Please provide breakdown of amount of monolingual data and bilingual data per language/language-pair. How many languages do you get monolingual data in? It's not clear from the paper.

- Also, please be explicit about the number of languages supported in the model (50). This is hidden in Appendix A. One can argue that this is not a fair comparison against XLM-R since it's trained on 100+ languages.

- Change "Ours implementation" to "Our implementation" everywhere.

Section 7:
- Change "targeting at initializing both..." to "targeted at ..."

---

> ### Author Response · Authors · 2020-11-19
> **Response to AnonReviewer1**
>
> Thank you for the comments! We address some of the concerns and questions below:
>
> **1-1: More details about datasets and languages.**
>
> Good suggestions. In our updated PDF version, we’ve added the number of languages and the amount of monolingual examples in Section 3. And we’ve also added detailed statistics of monolingual data per language and bilingual data per language-pair in Table 5, 6.
>
> **1-2: "What would happen if VECO was trained on 100 languages from XLM-R? "**
>
> We are working on training VECO on 100 languages (it just takes lots of compute!). The performance of multilingual models relates to both model size and language number. For a fixed-sized model, the per-language capacity decreases as we increase the number of languages [1]. Overall, low-resource language performance can be improved by adding similar higher-resource languages during pre-training (known as positive transfer), and high-resource language performance suffers from this capacity dilution [2]. There is a trade-off between positive transfer and capacity dilution. Moreover, the overall performance of downstream tasks also relies on the number of test languages. We want to emphasize that XLM-R is trained on 100 languages while mBART is trained on 25 languages. So we choose a medium number between them to balance the comparison of them NLU and NLG tasks.
>
> **1-3: More ablation experiments where the amount of pre-trained monolingual and parallel data were reduced independently.**
>
> It's interesting to show the performance of reducing two types of datasets independently to 1/2, 1/4, 0, which is a research gap among existing cross-lingual works. Table 2 has done partially experiments, showing that the performance of reducing the parallel (bilingual) corpus to 0 is 7.1% lower than trained on the full parallel (bilingual) corpus. We did not do too detailed experiments on this in the submission because it is costly and time-consuming to train several multilingual models from scratch and we were limited for time and paper space. We will add this additional ablation study to the paper (such as in the small-sized setting). Thanks for your suggestion.
>
> Finally, we appreciate your careful and rigorous comments on this paper (e.g., the details of the training corpus&languages, the citation of XTREME sub-tasks, and so on). We have fixed these minor problems in the updated version. Thanks again for taking the time and energy to help us improve our paper.
>
> **References**
>
> [1] Conneau, Alexis, et al. "Unsupervised cross-lingual representation learning at scale." arXiv preprint arXiv:1911.02116 (2019).
>
> [2] Arivazhagan, Naveen, et al. "Massively multilingual neural machine translation in the wild: Findings and challenges." arXiv preprint arXiv:1907.05019 (2019).

---

> > ### Comment · AnonReviewer1 · 2020-11-24
> > **Reply to the author response**
> >
> > I would like to thank the authors for improving the draft based on my suggestions which includes adding details about monolingual and bilingual data used. I definitely think that this paper is worthy of being presented at the conference given the very strong empirical results on both encoder-only and encoder-decoder tasks. They were #1 on the XTREME leaderboard and the same model can be also used to perform generation tasks. This is an important advantage of the paper. I like their supplemental figure which explains this more clearly. The CS_MLM objective is novel and enables them to use it for encoder-only and encoder-decoder tasks.

---

### Official Review · AnonReviewer5 · 2020-11-06
**acknowledge the promising results**

**Rating:** 4
**Confidence:** 5

**Review:**

This submission integrates the encoder-only and encoder-decoder Transformer for both understanding and generation tasks through parameter sharing. The authors present a variable encoder-decoder pre-training approach to unify the two mainstreams in pre-training tasks. To be specific, the model shares the self-attention layer and feed-forward layer for both encoder and decoder, while the cross-attention layer is trained in an alternative way. The VECO approach delivers strong results on various cross-lingual tasks of XTREME benchmark for understanding tasks, and generation tasks for WMT14 En-Fr and WMT14 En-De.

Comments:
First of all, the VECO approach provides impressive results on the different benchmark datasets including both language understanding and generation. It outperforms the previous methods in a non-trivial margin, which demonstrates the effectiveness of the approach. This is deeply acknowledged. Also, the studies and the detailed statements are mostly enough to make evaluations.  Despite its effectiveness, I do have several concerns and problems:
* In terms of the technical contribution, the relation between VECO and previous works is hard to make a strong difference, for example, BART, UniLM. Though the authors mentioned different language pairs (multilingual), personally, I acknowledge the parameter sharing of self-attention and feed-forward layer between the encoder and decoder is the main difference, while this is also widely used in machine translation models. Therefore, I feel a little bit unsatisfied with the contributions.
* As for the shared pertaining method, it is a little bit confused about the parameter updating. For the CS-MLM loss, the decoder will reuse the hidden states of encoder output, are these hidden states fixed during the decoder update? Or they will still be updated to the encoder (I know the parameters are shared)? The implementation detail is not so clear. If the encoder continues updating, how can the GPU memory cost differ from the MT fine-tuning procedure?
* The results are compared in a clear way, however, the parameters are not clearly compared with previous works. It seems the 662M model size is bigger than previous works. This is not the main weakness.

Generally speaking, this work is good for benchmark tasks to achieve strong results or a good technical report. But for the ICLR conference, it may need more.

----------

Post-update:
Thanks to the authors for your response. I deeply acknowledge the promising results achieved from your work, which is impressive. I still feel hard about the contribution, though I also acknowledge the difference and it is an excellent practical trick.

---

> ### Author Response · Authors · 2020-11-19
> **Response to AnonReviewer5**
>
> Thank you for the comments! We address some of the concerns and questions below:
>
> **5-1: The difference between BART and UniLM, and the novelty of this paper.**
>
> There are several major differences between VECO and BART/UniLM, in addition to monolingual v.s multilingual.
> - UniLM, an encoder-only model, requires the inputs $(x, y)$ to interact from the first layer, which may be suboptimal under the “multilingual” NLG scenario. Like machine translation, a better way is to first extract the “global” representation of the source language $x$ in the encoder and then interact with the representation of the target language $y$ in the decoder. UniLM lacks a global understanding of $x$.
> - BART and its multilingual version mBART, an encoder-decoder model, can be naturally used in NLG tasks. However, when applied to NLU tasks, BART feeds the same redundant sequence $x$ into the encoder and decoder and the final output in the decoder is used. The redundant input $x$ may lead BART to greater reliance on unidirectional language modeling in the decoder, while ignoring the bidirectional understanding of the input text in the encoder. Experiments on NLU benchmarks also show that BART can't beat RoBERTa, even with more parameters (i.e., the extra cross-attention) than the encoder-only model.
>
> **To sum up**, both the encoder-only model and encoder-decoder model need to be weighed against NLU and NLG downstream tasks in the *same parameters and fixed structure*. Thus, we propose VECO, a model that has a better trade-off between NLU and NLG in a view of *flexible parameter/structure*. The sharing of model parameters is just to achieve this goal of flexibility with the minimum parameters. We’ve tried to only share the same modules of mBART and trained it with denoised pre-training task, but finally got negative results in our early experiments. Thus, how to design reasonable pre-training tasks to let NLU and NLG boost each other under structural sharing, is another main contribution of this paper. The followings highlight the subtleties of the proposed pre-training tasks IS-MLM and CS-MLM:
> - IS-MLM: Force the model to bidirectionally comprehend of source input, which is a shortage of BART and MMTE since they don’t have any direct constraint (loss) in the encoder.  This task also guarantees the effectiveness of only extracting the "virtual" encoder of VECO (while dropping the "virtual" decoder) for NLU tasks.
> - CS-MLM: 1) Ensure the model to have a “global” bidirectional understanding of source text $x$ via cross-attention module, while UniLM lacks it; 2) Predict the masked words other than generating the next word, thus keeping in line with IS-MLM towards a more consistent optimization direction (predicting masked words) on the shared parameters. Thus, the CS-MLM task is totally different from the “shifted” generation task of MASS and BART.
>
> Although the two pre-training tasks seem *simple* and may easily be *confused* with other works at the first look, **the core contribution** of this article is the exquisite cooperation of parameters sharing and pre-training strategy makes it flexibly initialize any framework. We will re-organize some statements in our updated version that can let readers focus more on this point.
>
> **5-2: "Are the hidden states of encoder output fixed during the decoder update? how can the GPU memory cost differ from the MT fine-tuning procedure?"**
>
> The “hidden states” are fixed since they have computed in the forward process. I guess that you probably wonder whether the “parameters” related to the hidden states (e.g, self-attention and feed-forward) are fixed or not. When optimizing the CS-MLM objective, we “detach” the hidden states $X$ from the encoder to speed up and stabilize training. To be specific, it refers to “$X$.detach()” operation in Pytorch or the “tf.stop_gradient($X$)” operation in TensorFlow. That is to say, we pretend that the hidden states are a constant (like external knowledge). However, the self-attention and feed-forward parameters in the decoder will be updated during the back-propagation process. Then, in the next training step, the encoder will forward using the updated self-attention and feed-forward parameters. During fine-tuning on MT, we can 1) continue to tie parameters among encoder and decoder -> the same memory cost as pre-training, or 2) untie them with just the same values at the beginning but differ in later fine-tuning -> twice memory cost as pre-training.
>
> **5-3: “It seems the 662M model size is bigger than previous works.”**
>
> The sizes of pre-trained cross-lingual models are: XLM-R (550M) < VECO (662M) < mBART (680M)
>
> Regrading of pre-trained models, mBART has more parameters than VECO. When fine-tuning on NLU datasets, VECO has the same parameters as XLM-R since it drops the cross-attention modules. We have added Table 8 in the updated version to show the detailed comparisons in terms of model size and architecture, etc.

---

### Author Response · Authors · 2020-11-20
**Paper Updates**

We want to thank the reviewers again for their suggestions! We have updated the paper with the following changes:
- Addressing AnonReviewer1’s concern,  we added detailed statistics of  1) monolingual data per language in Table 5;  2) bilingual data per language-pair in Table 6. We are in the process of training VECO on 100 languages from scratch, although it is still early along in training.
- Addressing AnonReviewer2’s question, we added more descriptions (in Section 5.1) about why we use a different layer setting in MT experiments
- Addressing AnonReviewer3’s question, we 1) added a comparison to BART in terms of model framework and pre-training tasks in the end of Section 2.2;  2) added the reason why we flip the monolingual sentence order in Eq.(8); 3) fixed minor mistakes in Eq. (2).
- Addressing AnonReviewer4’s question, we 1) added the MMTE baseline in Table1-->13.6% lower than VECO; 2) provided more details about the computation and memory cost of utilizing BART/mBART for NLU downstream tasks in Section 5.1 to better clarify our claims.
- Addressing AnonReviewer5’s question, we 1) added some in-depth analysis at the end of Section 2.2 to clarify the novelty of VECO; 2) listed the other comparisons (e.g., parameters, architecture) between VECO and baselines in Table 8.

---

### Decision · Program_Chairs · 2021-01-07
**Final Decision**

**Decision:**

Reject

**Comment:**

This paper proposes a unified cross-lingual pretraining method that works well for both natural language understanding (NLU)—typically done using encoder-only architectures like mBERT and XLM—and conditional natural language generation (NLG) tasks like machine translation—typically done using encoder-decoder architectures like mBART.

This paper clearly split reviewers, with 2 quite or very positive on it, and 3 thinking or leaning towards thinking that it didn't have enough novelty to merit publication.

Pro
- The model produces good SoTA results
- The method is easily replicable
- It is good for the community for leading systems on benchmark tasks to have published papers describing how they work.

Con

- The work is not groundbreaking in technical novelty
- The work has to do a better job of communicating its contributions: It's hard to understand how it differs from other methods

On balance, the overall assessment is that the paper is not yet ready in its current form. The hope is that authors find the reviewer comments useful for preparing a future submission:

- The paper **has** to do a better job of communicating its contributions. All that most researchers got from the first version was that there was parameter sharing and that helped. The revised version starts to do a better job of explaining the value of having the IS-MLM and CS-MLM objectives to doing well on NLU and NLG tasks, but much more is needed, as the discussion here shows. Indeed, even the discussion here is often opaque. In describing the key contribution of the paper, in both the revised paper and discussion, the authors fall back on phrases like "elaborately designed" and "exquisite cooperation of parameter sharing and pre-training tasks". **What do "elaborately designed" and "exquisite cooperation" mean?!?** I think you can minimally clearly explain the benefits of having an objective like IS-MLM for doing better on NLU tasks than the approach taken in mBART. You could argue for the advantages of MLM vs LM generation, which has been shown in other papers, including the original BERT paper and ELECTRA. Concretely, I wonder if you should reverse the contents of section 2 and start with equation (8) and explain why that is a good objective for your system, and better than ones that have been used previously. This discussion should be at a higher level than the current discussion under (8) which tends to be in the weeds. I haven't worked all the details, but I think you could then describe the objectives of section 2.2 before describing the implementation in section 2.1, and the result might be clearer? It would certainly emphasize the importance of these loss functions.
- The initial version didn't have important details like the number of languages covered in the main paper; the current version fixes this to the extent of saying you have 50, but still doesn't give the context of how this compares with XLM-R and mBART. And several reviewers had questions about the number of parameters of different models. I think you could fix a lot of these concerns by moving Table 8 to the main paper in a future resubmission. It doesn't take up much space and helps a lot in providing these details and easy to find citations for the models compared in other papers.